# EDIT: Early Diffusion Inference Termination for dLLMs Based on the Dynamics of Training Gradients

## Abstract

Diffusion-based large language models (dLLMs) generate tokens through iterative denoising, but answers often stabilize before all denoising steps are completed. We introduce EDIT (Early Diffusion Inference Termination), an inference-time method that adaptively stops the denoising process once reasoning stability relative to training behavior is detected. EDIT is built on training-gradient dynamics, typically otherwise discarded after training, where, during fine-tuning, AdamW-aggregated LoRA updates encode parameter importance signals. We retain this information as compact reasoning maps. During inference, EDIT measures alignment between token activations and these maps, detecting convergence when KL divergence across consecutive steps on unmasked (visible) tokens falls below a threshold. On reasoning benchmarks, EDIT reduces diffusion steps by 11.8–68.3% while preserving or improving accuracy in most cases, with negligible storage overhead ($\sim$0.02%, about 1.5–2 MB for all QKV modules in a 32-block, 8 GByte model). These results establish a principled mechanism for transforming knowledge about training-gradient dynamics into practical test-time benefits such as reducing reasoning time.

## 1 Introduction

Modern language model deployment follows a wasteful paradigm: during training, optimization dynamics on gradients generate rich information about which parameters are critical for specific capabilities, yet this metadata is routinely discarded once training completes. We challenge this practice by demonstrating that training-time optimization trajectories contain valuable signals that can guide intelligent inference-time decisions, specifically enabling adaptive early inference termination for diffusion language models.

Diffusion-based language models (dLLMs) (Nie et al., 2025b; Zhao et al., 2025; Arriola et al., 2025) represent a promising alternative to autoregressive token generation, employing iterative denoising processes that progressively refine output. However, inference remains computationally expensive because of the fixed number of denoising steps, even when high-quality output emerges early in the process. Approaches such as Block Diffusion (Arriola et al., 2025) improve efficiency, but inference in dLLMs still relies on predetermined denoising schedules rather than terminating adaptively. As a result, current methods operate without awareness of which model parameters drove learning during training, leading to uninformed termination decisions. Efficiency in autoregressive LLMs has also been pursued through compression and restructuring, such as distillation (Sanh et al., 2019; Jiao et al., 2020; Gu et al., 2024), quantization (Frantar et al., 2022; Xiao et al., 2023; Dettmers et al., 2023; Lin et al., 2024; Dettmers et al., 2024), and pruning (Sanh et al., 2020; Frantar & Alistarh, 2023), yet these methods modify the model itself. EDIT instead reuses training-gradient information to adaptively determine when to stop inference, offering a different route to efficiency without modifying the model.

We introduce EDIT (Early Diffusion Inference Termination), a method that utilizes training optimization metadata to identify opportunities for early inference termination. Our key insight is that the AdamW optimizer's (Loshchilov & Hutter, 2019) moment estimates during fine-tuning training encode which parameters consistently receive strong, directionally-aligned updates when learning

reasoning tasks. Adaptive computation methods (Graves, 2016; Dehghani et al., 2019; Xin et al., 2020) have explored halting-based depth control, but these rely only on inference-time signals, without leveraging training-gradient dynamics. The effectiveness of EDIT is validated from the perspective of gradient convergence during inference, as shown in Figure 2. These patterns, which we term *AdamW evolution*, represent a map of the model's learned reasoning pathways. Rather than discarding this information when training is complete, we store it as compact metadata (requiring minimal additional storage) and use it to guide inference termination decisions.

**Our Approach: Utilizing Training Metadata for Guiding Inference Termination.** During supervised fine-tuning (SFT) on reasoning tasks, certain LoRA parameters receive consistent gradient signals across training steps, indicating their importance in encoding reasoning patterns. Evidence from pruning supports this view, as update-based criteria such as Movement Pruning (Sanh et al., 2020) also demonstrate that parameter importance can be inferred from gradient dynamics. The AdamW optimizer naturally tracks this through its moment estimates, with the first moment capturing gradient direction and the second moment reflecting gradient stability. Parameters with large, stable updates become critical components of the learned reasoning pathways. Traditional inference deployment discards this valuable information. EDIT preserves it by saving aggregated AdamW updates across training steps, creating a fingerprint of which parameters matter most for reasoning. At inference time, we compare current token activations against these preserved patterns using cosine similarity, assessing whether the model's current state aligns with its learned reasoning pathways. When this alignment stabilizes—indicating the model has reached its learned reasoning configuration—we can confidently terminate the diffusion inference process early. Instead of relying on inference-time heuristics such as confidence or output stability, EDIT leverages training-time knowledge to terminate once key reasoning components are engaged.

**Contributions.** (1) We establish a new paradigm for early inference termination that leverages training metadata, which is usually discarded in prior methods. (This approach opens future research directions not only in early inference termination, but also informing dynamic compute allocation, quality prediction, and other inference-time optimizations.) (2) We provide a practical instantiation through EDIT, demonstrating that AdamW evolution patterns can reliably indicate when diffusion models have completed their core reasoning. Our method requires no architectural changes, adds minimal storage overhead, and integrates seamlessly with existing diffusion language models. (3) We validate our approach on multiple reasoning benchmarks, showing inference speedups of 11.8% to 68.3% while maintaining or improving accuracy in most settings. These gains come purely from utilizing training information that already existed but was previously thrown away, highlighting the inefficiency of current practices.

## 2 PRESERVING AND UTILIZING TRAINING METADATA

We detail how EDIT captures optimization dynamics during training and leverages them for intelligent early termination during inference. Our approach consists of two phases: metadata extraction during fine-tuning (Section 2.1) and metadata-guided termination during inference (Section 2.2).

**Utilizing Training Metadata for Guiding Inference Termination.** During supervised fine-tuning (SFT), some parameters receive strong, stable updates that encode core reasoning patterns, while others show weak or oscillating updates and contribute less. We track this distinction through the AdamW update history—what we call the *AdamW evolution*—which forms a map of reasoning-relevant parameters. At inference, activations are compared against this map; when alignment with the learned reasoning pathways is reached, early termination is enabled with confidence.

### 2.1 CAPTURING ADAMW EVOLUTION DURING TRAINING

We consider a pre-trained base model with LoRA (Low-Rank Adaptation) (Hu et al., 2022) modules inserted into the Query ($Q$), Key ($K$), and Value ($V$) projections of each Transformer block. During SFT on reasoning tasks, only these LoRA parameters are updated. Each LoRA module consists of matrices $(A, B)$ where $A \in \mathbb{R}^{r \times d_{\text{in}}}$ and $B \in \mathbb{R}^{d_{\text{out}} \times r}$ implement a low-rank update to the corresponding projection.

**Notation note:** We use $\mathcal{L}_k$ for the loss at training step $k$ and $L$ for block length in the diffusion process, and we focus on LoRA-B matrices, as our ablation (Table 3) and visualization analysis (Appendix C.2) show they provide more stable and informative signals than LoRA-A.

For the LoRA-B matrix $B \in \mathbb{R}^{d_{out} \times r}$ (where $d_{out}$ is the output dimension and $r$ is the rank), the AdamW optimizer maintains first and second moment estimates at each training step $k$:

$$M_{k,B} = \beta_1 M_{k-1,B} + (1 - \beta_1)G_{k,B}, \quad V_{k,B} = \beta_2 V_{k-1,B} + (1 - \beta_2)G_{k,B}^{\odot 2}, \tag{1}$$

where $G_{k,B} = \nabla_B \mathcal{L}_k$ is the gradient tensor, $\beta_1, \beta_2 \in [0, 1)$ are decay rates, and $\odot$ denotes element-wise operations. The element-wise update magnitude at step $k$ is:

$$U_{k,B} = \frac{M_{k,B}}{\sqrt{V_{k,B}} + \epsilon}, \tag{2}$$

where $\epsilon$ ensures numerical stability and division is element-wise. To create a stable representation of the parameter importance patterns, we define the *AdamW evolution tensor* as the average over all $\mathcal{K}$ fine-tuning steps:

$$\bar{U}_B = \frac{1}{\mathcal{K}} \sum_{k=1}^{\mathcal{K}} U_{k,B} \in \mathbb{R}^{d_{out} \times r}. \tag{3}$$

This tensor captures which elements of the LoRA-B matrix consistently received strong, directional updates during training. To enable comparison with token activations $\mathbf{f}_s \in \mathbb{R}^{d_{out}}$, we reduce $\bar{U}_B$ to a feature-aligned vector $\mathbf{u} \in \mathbb{R}^d$ using row-wise energy:

$$\mathbf{u}[p] = \left\| \bar{U}_B[p, :] \right\|_2 = \sqrt{\sum_{j=1}^{r} \bar{U}_B[p, j]^2}, \quad p = 1, \ldots, d_{out}. \tag{4}$$

This reduction preserves the update magnitude each output dimension receives across low-rank components, forming a parameter-importance signature aligned with the feature space.

## 2.2 METADATA-GUIDED EARLY TERMINATION DURING INFERENCE

At inference time, we leverage the preserved AdamW evolution to determine when the diffusion process can safely terminate. Our approach operates on block-level diffusion (Nie et al., 2025b; Zhao et al., 2025; Arriola et al., 2025), where the sequence is divided into blocks of length $L$, and tokens within each block are progressively unmasked across denoising steps.

### 2.2.1 ASSESSING REASONING ALIGNMENT

Let $\mathcal{S}_t$ denote the set of visible (unmasked) tokens at denoising step $t$. For each visible token $s \in \mathcal{S}_t$, we extract its post-LoRA activation $\mathbf{f}_s^{(t)} \in \mathbb{R}^{d_{out}}$ from the chosen module (specifically, the LoRA-B output of the Query projection in the last Transformer block, based on our empirical findings in Figure 4). We compute the cosine similarity between each token's activation and the AdamW evolution vector:

$$\text{Sim}_s^{(t)} = \frac{\langle \mathbf{f}_s^{(t)}, \mathbf{u} \rangle}{\|\mathbf{f}_s^{(t)}\|_2 \|\mathbf{u}\|_2}, \tag{5}$$

where $\mathbf{u}$ is the feature-aligned vector from Equation 4. To convert these alignment scores into a probability distribution, we apply softmax with a fixed temperature $\tau_{\text{blk}}$ within each block:

$$P^{(t)}(s) = \frac{\exp(\text{Sim}_s^{(t)} / \tau_{\text{blk}})}{\sum_{i \in \mathcal{S}_t} \exp(\text{Sim}_i^{(t)} / \tau_{\text{blk}})}, \quad s \in \mathcal{S}_t. \tag{6}$$

Keeping $\tau_{\text{blk}}$ fixed within a block ensures that distribution changes reflect genuine alignment shifts rather than temperature-induced artifacts.

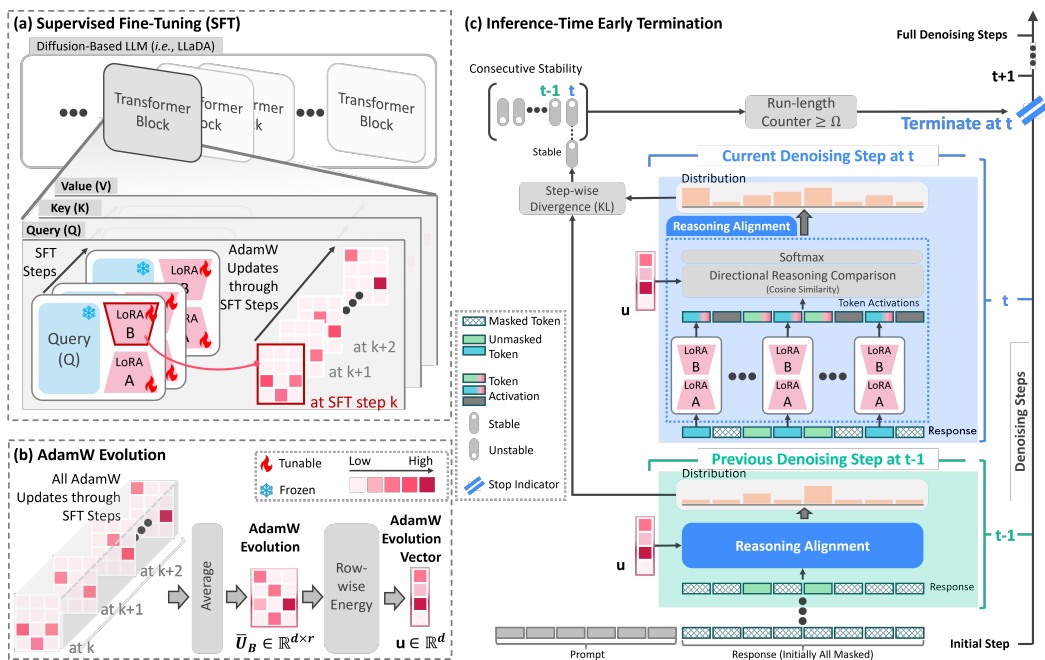

Figure 1: Overview of EDIT. *(a) Supervised Fine-Tuning (SFT)*: AdamW moment estimates track LoRA-B updates across steps, where some parameters consistently receive strong, directionally-aligned updates that encode reasoning patterns. *(b) AdamW Evolution*: Aggregating these updates yields a compact evolution vector **u** that encodes reasoning-relevant parameter importance. *(c) Inference-Time Early Termination*: At inference, token activations are compared with the preserved evolution vector **u**; reasoning alignment is monitored via cosine similarity and KL divergence across steps. Once stability persists for consecutive steps, termination occurs before full denoising, reducing cost without loss of quality.

### 2.2.2 DETECTING REASONING STABILITY VIA MATCHED SUPPORT

As tokens are progressively unmasked, the support of our distribution grows. To properly compare distributions across steps, we must account for this changing support. Let $\mathcal{I}_t = \mathcal{S}_{t-1} \cap \mathcal{S}_t$ be the intersection of visible tokens between consecutive steps. We renormalize both distributions to this common support:

$$\tilde{P}^{(t)}(s) = \frac{P^{(t)}(s)}{\sum_{i \in \mathcal{I}_t} P^{(t)}(i)}, \quad \tilde{P}^{(t-1)}(s) = \frac{P^{(t-1)}(s)}{\sum_{i \in \mathcal{I}_t} P^{(t-1)}(i)}, \quad s \in \mathcal{I}_t. \tag{7}$$

The step-wise divergence is then computed as:

$$D_t = D_{\text{KL}}(\tilde{P}^{(t)} \parallel \tilde{P}^{(t-1)}) = \sum_{s \in \mathcal{I}_t} \tilde{P}^{(t)}(s) \log \frac{\tilde{P}^{(t)}(s)}{\tilde{P}^{(t-1)}(s)}. \tag{8}$$

### 2.2.3 EARLY TERMINATION WITH CONSECUTIVE STABILITY

To ensure robust termination decisions, we require consecutive steps of stability rather than sporadic stable steps. We maintain a run-length counter $c$ updated as $c \leftarrow c + 1$ if $D_t < \delta$, and reset to 0 otherwise. The diffusion process for the current block terminates when $c \geq \Omega$, indicating that the model's reasoning alignment has remained stable for $\Omega$ consecutive steps. Figure 1 and Algorithm 1 present the full EDIT workflow and its procedure.

Formally, if the divergence $D_t$ remains below $\delta$ for $\Omega$ consecutive steps, then the cumulative drift between step distributions is bounded by

$$\text{TV}(\tilde{P}^{(t)}, \tilde{P}^{(t-\Omega)}) \leq \Omega\sqrt{\tfrac{\delta}{2}}, \tag{9}$$

where $\mathrm{TV}(p,q) = \frac{1}{2}\sum_i |p_i - q_i|$ denotes the total variation distance (Lemma D.1). Moreover, if this bound is smaller than half of the prediction margin

$$\Omega\sqrt{\tfrac{\delta}{2}} < \tfrac{1}{2}m_t, \tag{10}$$

with $m_t$ the gap between the highest and second-highest probabilities at step t, then the predicted token cannot change during the stability window (Theorem D.2).

These conditions form the basis of EDIT's termination rule. When the matched-support KL divergence remains small over $\Omega$ consecutive steps, where $\Omega$ is determined by the inference progress at the point of early termination, the cumulative variation is bounded, which keeps predictions stable, prevents token flips, and controls Lipschitz observables. PAC-style bounds (Corollary D.6) provide a principled way to select $(\delta, \Omega)$ and show that early termination matches full denoising with high probability. Appendix D gives full statements and calibration rules, while Appendix E and Appendix F present extensions on token-wise freezing with instance-level safety and subspace formulations that preserve the same properties.

The training phase extracts metadata with zero additional computational cost (these values are already computed by the optimizer), while the inference phase uses this metadata to make principled termination decisions.

**EDIT Algorithm.** The complete procedure of EDIT is summarized in Algorithm 1, which outlines both the training-time extraction of metadata and the inference-time termination rule.

---

**Algorithm 1** EDIT: Early Diffusion Inference Termination

---

**Require:** Input sequence; thresholds $\delta$, $\Omega$; block temperature $\tau_{\mathrm{blk}}$; fine-tuning steps $\mathcal{K}$
**Ensure:** Generated text with adaptive early termination
 1: **// Phase 1: Training-Time Metadata Extraction (one-time)**
 2: **for** each fine-tuning step $k = 1$ to $\mathcal{K}$ **do**
 3:     Update AdamW moments $M_{k,B}, V_{k,B}$ using Eq. 1
 4:     Compute update tensor $U_{k,B}$ using Eq. 2
 5: **end for**
 6: Compute AdamW evolution tensor $\bar{U}_B$ using Eq. 3
 7: Reduce to feature vector $\mathbf{u}$ using Eq. 4
 8: Store $\mathbf{u}$ as metadata for inference use
 9: **// Phase 2: Inference-Time Early Termination (uses precomputed $\mathbf{u}$)**
10: **for** each diffusion block $b = 1$ to $B$ **do**
11:     Initialize visible set $\mathcal{S}_1$ by unmasking schedule
12:     Compute activations $\mathbf{f}_s^{(1)}$ for $s \in \mathcal{S}_1$
13:     Compute initial distribution $P^{(1)}$ using Eq. 5 and 6
14:     Set stability counter $c \leftarrow 0$
15:     **for** denoising step $t = 2$ to $T_b$ **do**
16:         Update visible set $\mathcal{S}_t$ according to unmasking schedule
17:         Compute activations $\mathbf{f}_s^{(t)}$ for $s \in \mathcal{S}_t$
18:         Compute distribution $P^{(t)}$ using Eq. 5 and 6
19:         Set intersection $\mathcal{I}_t = \mathcal{S}_{t-1} \cap \mathcal{S}_t$
20:         Renormalize to $\tilde{P}^{(t)}, \tilde{P}^{(t-1)}$ using Eq. 7
21:         Compute $D_t = D_{\mathrm{KL}}(\tilde{P}^{(t)} \parallel \tilde{P}^{(t-1)})$ using Eq. 8
22:         **if** $D_t < \delta$ **then**
23:             $c \leftarrow c + 1$
24:         **else**
25:             $c \leftarrow 0$
26:         **end if**
27:         **if** $c \geq \Omega$ **then**
28:             **break** // Early termination for block $b$
29:         **end if**
30:     **end for**
31: **end for**

---

Table 1: Accuracy on reasoning benchmarks. EDIT uses training-time metadata for adaptive early termination. Results are mean over 3 seeds, where **bold** denotes the best, underline denotes the second-best. 0-shot means no in-context examples during evaluation (post-SFT). Experiments are run with sequence lengths 128/256/512. The symbol ($^{\dagger}$) indicates results reproduced from (Zhao et al., 2025) with Intel XPU hardware.

| Dataset (Seq Len) Method | Countdown (0-shot) | | | Sudoku (0-shot) | | | MATH500 (0-shot) | | | GSM8K (0-shot) | | | GPQA (0-shot) | | |
|---|---|---|---|---|---|---|---|---|---|---|---|---|---|---|---|
| | 128 | 256 | 512 | 128 | 256 | 512 | 128 | 256 | 512 | 128 | 256 | 512 | 128 | 256 | 512 |
| LLaDA (No SFT)$^{\dagger}$ (Zhao et al., 2025) | 19.9 | 19.5 | 16.4 | 10.4 | 6.4 | 6.3 | **27.6** | 32.4 | 36.0 | 68.1 | 75.8 | 79.5 | 21.9 | **27.9** | 25.7 |
| LLaDA (SFT)$^{\dagger}$ (Zhao et al., 2025) | 19.5 | 20.7 | 20.3 | 11.4 | 8.2 | 5.0 | 26.2 | 30.4 | 35.4 | **69.8** | 77.0 | **81.2** | 23.0 | 20.5 | **26.3** |
| **EDIT (Ours)** | **28.9** | **31.6** | **27.7** | **16.1** | **11.3** | 7.6 | 27.4 | **32.8** | **36.6** | 67.3 | **77.6** | 76.2 | **25.5** | 27.7 | 26.1 |

Table 2: Diffusion steps with EDIT vs. baseline full diffusion. Values are averaged across blocks, and baselines are fixed at 64/128/256 steps for sequence lengths 128/256/512. Percentages show reduction from baseline steps, with training metadata enabling confident early termination without quality loss.

| Dataset (Seq Len) Method | Countdown | | | Sudoku | | | MATH500 | | | GSM8K | | | GPQA | | |
|---|---|---|---|---|---|---|---|---|---|---|---|---|---|---|---|
| | 128 | 256 | 512 | 128 | 256 | 512 | 128 | 256 | 512 | 128 | 256 | 512 | 128 | 256 | 512 |
| Baseline (Full Steps) | 64 | 128 | 256 | 64 | 128 | 256 | 64 | 128 | 256 | 64 | 128 | 256 | 64 | 128 | 256 |
| **EDIT (Ours)** | 40.4 | 40.6 | 133.3 | 38.3 | 74.9 | 163.3 | 38.1 | 81.9 | 197.2 | 42.8 | 103.5 | 225.8 | 40.3 | 81.3 | 194.1 |
| **Reduction (%)** | 36.9 | 68.3 | 47.9 | 40.2 | 41.5 | 36.2 | 40.5 | 36.0 | 23.0 | 33.1 | 19.2 | 11.8 | 37.0 | 36.5 | 24.2 |

## 3 EXPERIMENTAL VALIDATION

### 3.1 EXPERIMENTAL SETUP

We evaluate on five reasoning tasks: Countdown (Pan et al., 2025), Sudoku (Arel, 2025), MATH500 (Lightman et al., 2024), GSM8K (Cobbe et al., 2021), and GPQA (Rein et al., 2023). We use LLaDA-8B (Nie et al., 2025b) as our baseline model, fine-tuned on the s1K dataset (Muennighoff et al., 2025) with LoRA applied to QKV projections. All experiments use Intel XPU hardware to ensure reproducibility.

During SFT, we preserve AdamW evolution metadata (Section 2.1). Persisting reduced vectors $\mathbf{u}$ requires ∼16 KB per module, or ∼1.5 MB for all QKV projections across 32 Transformer blocks (<0.02% of an 8 GB model). At inference, EDIT uses task-specific thresholds (Appendix 3.2) selected on held-out validation sets (20% of training data), ensuring no test set leakage.

### 3.2 HYPERPARAMETER SELECTION PROTOCOL

To ensure reproducibility and avoid overfitting, we employ a systematic hyperparameter selection protocol. For each task, we use 20% of the training data as a validation set to tune the stability threshold $\delta \in \{0.025, 0.05, 0.1, 0.25, 0.45, 0.55\}$ and stability span $\Omega \in \{6, 8, 10, 12\}$. We select the configuration that maximizes the accuracy-efficiency trade-off, defined as accuracy divided by average diffusion steps. The block temperature $\tau_{\mathrm{blk}}$ is fixed at 1.0 for all experiments to ensure fair comparison. Table 5 in Appendix C.1 provides the complete configuration for each benchmark.

### 3.3 RESULTS: EFFICIENCY GAINS WITH PRESERVED ACCURACY

Table 1 shows that EDIT improves accuracy on Countdown (up to 31.6%) and Sudoku (up to 16.1%), while remaining competitive on other tasks. These gains arise because early termination avoids late-step degradation. Once predictions stabilize, further denoising can overwrite correct intermediate states, an effect pronounced in tasks with crisp solutions such as Countdown and Sudoku. In contrast, GSM8K at sequence length 512 drops from 81.2% to 76.2%, reflecting long reasoning chains where stability is detected before the model finishes all reasoning steps. Results on GSM8K and GPQA underscore task-specific variation, but the overall average shows a net positive effect, validating that metadata-guided termination enhances rather than compromises quality. Table 2 shows that EDIT reduces average denoising steps per block by 11.8%–68.3% compared with baselines fixed at 64/128/256 for sequence lengths 128/256/512. Gains are most pronounced on shorter se-

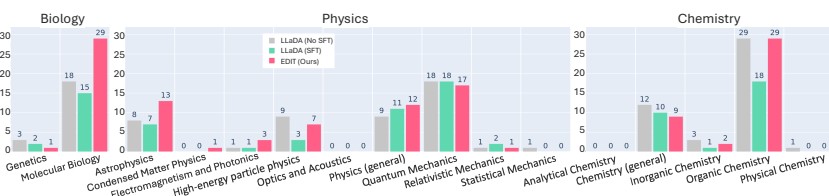

Figure 3: Performance breakdown across GPQA subdomains comparing EDIT (red) with baseline SFT (green). EDIT shows particularly strong improvements in Molecular Biology and Astrophysics, where reasoning patterns are more structured. The domain-specific variation validates that training metadata captures specialized reasoning pathways.

quences, where full diffusion is wasteful, yielding substantial computational savings. With PAC-style calibration (Appendix D.6), EDIT's termination decisions satisfy the correctness conditions of Corollary D.6 and often operate within certified safety bounds while delivering substantial speedups.

### 3.4 GRADIENT-BASED JUSTIFICATION FOR EARLY TERMINATION

To determine when denoising steps can be truncated safely, we adopt a gradient view of inference by comparing pseudo-gradients at inference with SFT gradients on LoRA-B layers. At each inference step $t$, the model outputs logits $z_t(s)$ for tokens $s$ in a block of length $L$. Since no ground-truth labels are available, the signal comes from changes in predictive distributions across steps. Let $p_\theta(z_t(s))$ and $p_\theta(z_{t+1}(s))$ denote predictions at steps $t$ and $t+1$; their KL divergence quantifies prediction change (see Appendix G for details). We define the pseudo-gradient as $\tilde{G}_{t,B} = \nabla_B \sum_{s \in S_{t+1}} \mathrm{KL}(p_\theta(z_t(s)) \| p_\theta(z_{t+1}(s)))$, where $S_{t+1}$ is the visible token set at step $t+1$. Back-propagating this divergence through LoRA-B yields $\tilde{G}_{t,B}$, whose root mean square (RMS) magnitude provides a scalar summary per step. Tracing these values across denoising steps produces a trajectory of inference dynamics.

During SFT, we compute RMS magnitudes of gradients $G_{k,B}$ across training steps and summarize them by a mean $\mu_{\mathrm{SFT}}$ and a variance band, defining the stable regime. Convergence is declared when inference pseudo-gradients (1) approach $\mu_{\mathrm{SFT}}$ and (2) remain within this band, beyond this time further denoising adds cost without benefit. For the

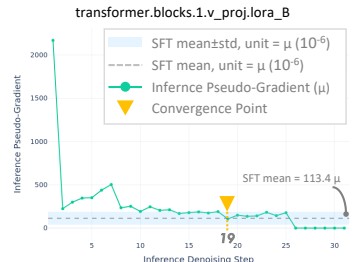

Figure 2: Gradient-based analysis of training–inference alignment on GPQA (sequence length 128, 2nd block). Root mean square (RMS) pseudo-gradients $\tilde{G}_{t,B}$ across steps are compared with the SFT gradient mean (dashed) and variance band (shaded). The convergence point (yellow ▼) occurs at step 19, after which pseudo-gradients stabilize near the SFT mean, showing that $\sim$20 steps per block preserve fidelity with lower computation (Table 2, 40.3 steps for two blocks).

GPQA benchmark (sequence length 128, 2nd block), this analysis is computed over all samples, and Figure 2 shows pseudo-gradients converging at step 19 (yellow ▼) (see more examples in Appendix G). Thereafter they fluctuate around the SFT mean, indicating entry into the training-consistent regime. Terminating at $\sim$20 steps per block thus preserves fidelity while cutting cost, consistent with Table 2 (40.3 steps for two blocks) and Table 1, which confirms accuracy remains competitive.

### 3.5 UNDERSTANDING WHEN TRAINING METADATA HELPS

Figure 3 shows that EDIT's effectiveness varies across problem types within GPQA. It yields large gains in domains requiring systematic reasoning (Molecular Biology, Astrophysics) while providing modest gains in others. This variation supports our core thesis: the training metadata captures task-specific patterns, and its utility depends on how well-defined these patterns are for each domain. Tasks with clear, consistent reasoning pathways benefit most from our approach.

Figure 4 visualizes how different reasoning tasks activate distinct parameter subsets, as revealed by the AdamW evolution patterns. In GPQA, for example, subdomains such as Astrophysics and Molecular Biology activate distinct subsets in the LoRA-B Query projection of the last Transformer

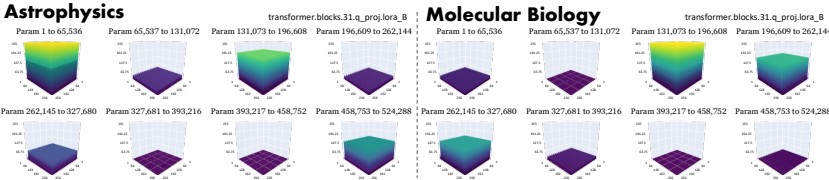

Figure 4: Task-dependent parameter importance in LoRA-B Query (transformer.block.31) revealed by AdamW evolution. GPQA subdomains (Astrophysics vs. Molecular Biology) emphasize different regions of the parameter space. The projection ($d_{out} = 4096, r = 128$) contains 524,288 parameters, reshaped into a $256 \times 256 \times 8$ grid, with the Z-axis showing normalized AdamW update magnitudes (0–255). The X–Y plane indexes parameters, and the Z-axis reflects activation strength, highlighting metadata-encoded reasoning pathways.

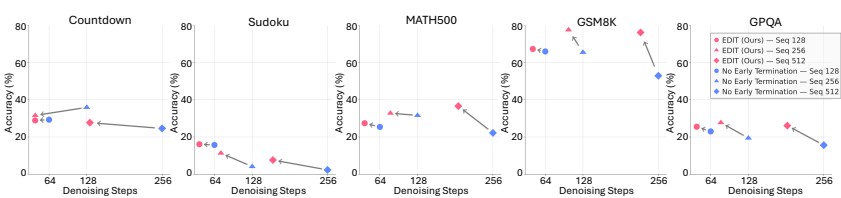

Figure 5: Ablation study of early termination effectiveness on reasoning benchmarks. EDIT with early termination (red) avoids the degradation observed with full denoising steps (blue). The x-axis represents the number of denoising steps, the y-axis represents accuracy, and the solid gray arrow highlights the improvement from full denoising to early termination.

block (block 31). This visualization shows that training dynamics produce meaningful signatures that can guide inference decisions.

## 3.6 EFFECTIVENESS OF EARLY TERMINATION AND MODULE SELECTION ANALYSIS

Figure 5 compares EDIT with early termination against its full-step counterpart. Across reasoning benchmarks, EDIT detects stability and typically achieves higher accuracy while requiring fewer denoising steps. The full-step counterpart, in contrast, continues past the stability plateau, often introducing fluctuations that do not enhance predictions. This demonstrates that early termination reduces computation and, in most cases, yields stronger results across tasks.

We compare LoRA-A and LoRA-B by analyzing feature-aligned deviation vectors of parameter changes during SFT across Transformer blocks, using four sparsity metrics (Appendix B). Top-$k$ Mass measures update concentration (lower indicates more spread), Participation Ratio (PR) (Martin & Mahoney, 2021) estimates the number of active parameters (higher indicates broader participation), Gini coefficient (Hurley & Rickard, 2009) captures inequality (lower indicates more even distribution), and Entropy (Huang & Tran, 2019) quantifies diversity (higher indicates greater spread). Across QKV projections, LoRA-B shows lower Top-$k$ Mass and Gini but higher PR and Entropy, reflecting broader, less sparse updates and a more stable training signal.

## 3.7 STORAGE AND COMPUTATIONAL OVERHEAD

The AdamW evolution metadata requires storing only the reduced vector $\mathbf{u} \in \mathbb{R}^d$ per chosen LoRA module, not the full tensor $\bar{U}_B$. For our configuration with $d = 4096$, this amounts to approximately 16 KB per module (assuming float32 precision). Even if we store metadata for all QKV projections across all 32 Transformer blocks, the total overhead is $32 \times 3 \times 16$ KB $\approx 1.5$ MB—merely 0.02% of the 8 GB model size.

At inference time, EDIT adds cosine similarity computations (Equation 5) and KL divergence calculations (Equation 8) at each denoising step. These operations have complexity $O(|\mathcal{S}_t| \cdot d)$ and $O(|\mathcal{I}_t|)$ respectively, which is minimal compared to the $O(L^2 \cdot d)$ cost of self-attention in each Transformer block. The net result is substantial speedup despite these additional computations.

Table 3: Sparsity metrics for LoRA-A and LoRA-B, averaged over 32 Transformer blocks. Reported for Query ($Q$), Key ($K$), and Value ($V$) projections with Top-$k$ Mass at $k = 30\%$. LoRA-B shows lower sparsity and more distributed updates. **Bold** numbers indicate the better value per metric.

| | Query ($Q$) | | | | Key ($K$) | | | | Value ($V$) | | | |
|---|---|---|---|---|---|---|---|---|---|---|---|---|
| Matrix | Top-$k$ Mass ($\downarrow$) | PR ($\uparrow$) | Gini ($\downarrow$) | Entropy ($\uparrow$) | Top-$k$ Mass ($\downarrow$) | PR ($\uparrow$) | Gini ($\downarrow$) | Entropy ($\uparrow$) | Top-$k$ Mass ($\downarrow$) | PR ($\uparrow$) | Gini ($\downarrow$) | Entropy ($\uparrow$) |
| LoRA-A | 0.3492 | 3764.3338 | 0.0782 | 0.9979 | 0.3397 | 3839.3339 | 0.0645 | 0.9981 | 0.3542 | 3744.9923 | 0.0871 | 0.9982 |
| LoRA-B | **0.3250** | **4007.5958** | **0.0389** | **0.9997** | **0.3209** | **4027.5595** | **0.0330** | **0.9998** | **0.3431** | **3857.9129** | **0.0652** | **0.9992** |

## 4 RELATED WORK

**Diffusion Language Models.** Diffusion-based large language models (dLLMs) progressed from continuous latent formulations for discreteness (Gulrajani & Hashimoto, 2023; Austin et al., 2021) to discrete and masked approaches that scale to larger architectures (Nie et al., 2025a; Ou et al., 2025; Shi et al., 2024; Sahoo et al., 2024; Gong et al., 2025; Nie et al., 2025b; Zhao et al., 2025; Arriola et al., 2025). They perform well on reasoning tasks such as chain-of-thought generation (Ye et al., 2024) and reversal (Ye et al., 2025a), and hybrid methods like block-wise diffusion further improve efficiency (Arriola et al., 2025). Large-scale systems show that dLLMs (Nie et al., 2025b; Ye et al., 2025b) approach autoregressive baselines, yet inference still relies on fixed denoising schedules that run all steps even when answers stabilize early. EDIT introduces adaptive termination from training-gradient dynamics to stop once stability is reached, avoiding redundant computation.

**Adaptive Computation.** Adaptive computation has been studied through halting-based depth mechanisms (Graves, 2016; Dehghani et al., 2019) and later applied to Transformers with early exiting and depth-adaptive methods (Xin et al., 2020; Elbayad et al., 2020; Liu et al., 2020; Xin et al., 2021). Confidence-based halting reduced cost (Schuster et al., 2021; 2022), and subsequent work explored training-free adaptation, pruning, and routing (Storaï et al., 2025; Lin et al., 2025; Salehi et al., 2023). These approaches adapt inference using heuristics such as entropy or confidence. EDIT instead uses training-gradient dynamics, with AdamW-aggregated updates acting as compact reasoning maps that provide reliable signals for deciding when to stop.

**Inference Efficiency.** Inference efficiency in LLMs has relied on compression and restructuring, including distillation (Sanh et al., 2019; Jiao et al., 2020; Gu et al., 2024), quantization (Frantar et al., 2022; Xiao et al., 2023; Dettmers et al., 2023; Lin et al., 2024; Dettmers et al., 2024), pruning (Wang et al., 2020; Frantar & Alistarh, 2023; Ma et al., 2024; Gao et al., 2024; Ashkboos et al., 2024; Zhong et al., 2025; Chen et al., 2025), and speculative inference (Leviathan et al., 2023; Miao et al., 2024). These approaches modify parameters or architectures to shrink models or reduce decoding cost. EDIT instead serves as a lightweight wrapper that leverages training-gradient metadata to terminate inference early, achieving efficiency without altering the model.

## 5 CONCLUSION AND FUTURE DIRECTIONS

We introduced EDIT, which preserves training metadata typically discarded in previous works and leverages it to guide early inference termination in diffusion language models. By capturing optimization dynamics during fine-tuning, EDIT signals when reasoning is complete, reducing inference cost without architectural changes. Across five reasoning benchmarks, it achieves 11.8–68.3% fewer diffusion steps while maintaining or improving accuracy, with only 0.02% storage overhead.

EDIT has limitations: it requires training dynamics, often unavailable in released models, suggesting providers include optimization metadata; it depends on task-specific thresholds $(\delta, \Omega)$, motivating adaptive or meta-learned criteria; and it is evaluated only with LoRA fine-tuning, leaving full-parameter extensions as future work. Beyond early termination, utilizing training meta data during inference could enable dynamic compute allocation across layers, guide token-level processing, predict generation quality before completion, and identify which prompts will benefit most from additional computation. More broadly, this work exposes a systemic inefficiency: training information is often discarded even though it can strengthen inference. Utilizing training metadata for guiding inference offers a path toward more holistic and efficient machine learning pipelines.

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

# Appendix

## A  MATHEMATICAL DETAILS

### A.1  COMPLETE DERIVATION OF ADAMW EVOLUTION

We provide the complete mathematical framework for extracting and utilizing AdamW evolution patterns. The key insight is that optimization dynamics during training create signatures of parameter importance that can guide inference.

For a LoRA-B matrix $B \in \mathbb{R}^{d_{out} \times r}$, the AdamW optimizer maintains exponentially weighted moving averages of gradients and squared gradients. Starting from the recursive definitions:

$$M_{k,B}[i,j] = \beta_1 M_{k-1,B}[i,j] + (1 - \beta_1) G_{k,B}[i,j] \tag{11}$$

$$V_{k,B}[i,j] = \beta_2 V_{k-1,B}[i,j] + (1 - \beta_2) G_{k,B}[i,j]^2 \tag{12}$$

By unrolling these recursions and assuming zero initialization, we obtain:

$$M_{k,B}[i,j] = (1-\beta_1)\sum_{\ell=1}^{k}\beta_1^{k-\ell}G_{\ell,B}[i,j] \tag{13}$$

$$V_{k,B}[i,j] = (1-\beta_2)\sum_{\ell=1}^{k}\beta_2^{k-\ell}G_{\ell,B}[i,j]^2 \tag{14}$$

The element-wise update magnitude captures both direction (through $M$) and reliability (through $V$):

$$U_{k,B}[i,j] = \frac{M_{k,B}[i,j]}{\sqrt{V_{k,B}[i,j]}+\epsilon} \tag{15}$$

Averaging across all training steps yields the AdamW evolution tensor:

$$\bar{U}_B[i,j] = \frac{1}{\mathcal{K}}\sum_{k=1}^{\mathcal{K}}U_{k,B}[i,j] \tag{16}$$

The reduction to feature space via row-wise energy (Equation 4) preserves the total update magnitude each output dimension received, creating an interpretable signature of parameter importance.

## A.2 THEORETICAL JUSTIFICATION FOR STABILITY DETECTION

The KL divergence between consecutive alignment distributions provides a principled measure of reasoning stability. Under mild assumptions about the smoothness of the denoising process, we can show that stable KL divergence indicates convergence to a fixed point in the alignment space.

Consider the alignment distribution as a function of the denoising step: $P^{(t)} = f_t(\mathbf{x},\mathbf{u})$ where $\mathbf{x}$ represents the current token states and $\mathbf{u}$ is the fixed AdamW evolution vector. If the denoising process is contractive in the alignment space (which can occur under conditions such as Lipschitz continuity of the denoiser combined with fixed-temperature softmax normalization), then:

$$D_{\text{KL}}(P^{(t+1)} \parallel P^{(t)}) \leq \gamma \cdot D_{\text{KL}}(P^{(t)} \parallel P^{(t-1)}) \tag{17}$$

for some $\gamma < 1$. This ensures that requiring $D_t < \delta$ for $\Omega$ consecutive steps provides strong evidence of convergence.

## B ANALYZING PARAMETER CHANGE DYNAMICS IN LORA-A VS. LORA-B

To decide whether LoRA-A or LoRA-B is the preferable parameterization, we analyze how strongly their parameters change during SFT. For each of the $\mathcal{K}$ finetuning steps, the AdamW optimizer produces element-wise update tensors $U_{k,A} \in \mathbb{R}^{r \times d_{\text{in}}}$ and $U_{k,B} \in \mathbb{R}^{d_{\text{out}} \times r}$ (Equations 1–2). To obtain a stable measure of parameter change, we compute mean-absolute-deviation tensors:

$$\bar{U}_A = \tfrac{1}{\mathcal{K}}\sum_{k=1}^{\mathcal{K}}U_{k,A}, \qquad \hat{U}_A = \tfrac{1}{\mathcal{K}}\sum_{k=1}^{\mathcal{K}}\big|U_{k,A}-\bar{U}_A\big|, \tag{18}$$

$$\bar{U}_B = \tfrac{1}{\mathcal{K}}\sum_{k=1}^{\mathcal{K}}U_{k,B}, \qquad \hat{U}_B = \tfrac{1}{\mathcal{K}}\sum_{k=1}^{\mathcal{K}}\big|U_{k,B}-\bar{U}_B\big|. \tag{19}$$

We then reduce along the low-rank dimension to obtain *feature-aligned deviation vectors* $\hat{\mathbf{u}}_A \in \mathbb{R}^{d_{\text{in}}}$ and $\hat{\mathbf{u}}_B \in \mathbb{R}^{d_{\text{out}}}$:

$$\hat{\mathbf{u}}_A[p] = \big\|\hat{U}_A[:,p]\big\|_2, \quad p=1,\dots,d_{\text{in}}, \tag{20}$$

$$\hat{\mathbf{u}}_B[p] = \big\|\hat{U}_B[p,:]\big\|_2, \quad p=1,\dots,d_{\text{out}}. \tag{21}$$

Here, the index $p$ corresponds to a feature dimension of the $Q$, $K$, or $V$ projection where the LoRA module is applied. Each entry $\hat{\mathbf{u}}[p] \geq 0$ quantifies how much the parameters associated with that dimension consistently change across the entire SFT trajectory. Intuitively, a *dense* $\hat{\mathbf{u}}$ indicates broad parameter participation, whereas a *spiky* $\hat{\mathbf{u}}$ indicates sparse or localized adaptation. We note that the deviation form $\hat{\mathbf{u}}$ is used only for analysis, as it highlights stable change patterns and enables robust sparsity comparisons between LoRA-A and LoRA-B. In contrast, EDIT relies on the feature-aligned vector based on average updates, which provides a direct signal for termination decisions.

## B.1 SPARSITY METRICS

We compute the following metrics on both deviation vectors $\hat{\mathbf{u}}_A \in \mathbb{R}^{d_{\text{in}}}$ and $\hat{\mathbf{u}}_B \in \mathbb{R}^{d_{\text{out}}}$. For notational simplicity, we write the definitions using a generic $\hat{\mathbf{u}} \in \mathbb{R}^d$, where $d = d_{\text{in}}$ for LoRA-A and $d = d_{\text{out}}$ for LoRA-B. Each metric provides a complementary perspective on how concentrated or distributed the parameter updates are, and together they capture the sparsity structure of adaptation.

**Top-$k$ Mass ($\downarrow$).** Fraction of the update magnitude concentrated in the top $k\%$ of coordinates:

$$\text{Top-}k\text{ Mass}(\hat{\mathbf{u}}; k) = \frac{\sum_{p \in \text{Top-}k(\hat{\mathbf{u}})} \hat{\mathbf{u}}[p]}{\sum_{p=1}^{d_{\text{out}}} \hat{\mathbf{u}}[p]}. \tag{22}$$

Lower values indicate that updates are more evenly spread, while higher values signal strong concentration in a few parameters.

**Participation Ratio (PR, $\uparrow$).** Effective number of parameters that meaningfully contribute to updates (Martin & Mahoney, 2021):

$$\text{PR}(\hat{\mathbf{u}}) = \frac{\left( \sum_{p=1}^{d} \hat{\mathbf{u}}[p]^2 \right)^2}{\sum_{p=1}^{d} \hat{\mathbf{u}}[p]^4}. \tag{23}$$

Low PR corresponds to sparse adaptation dominated by a few dimensions, while high PR indicates that many dimensions jointly participate.

**Gini Coefficient ($\downarrow$).** Measures inequality of participation (Hurley & Rickard, 2009). For sorted entries $x_1 \leq \cdots \leq x_d$:

$$\text{Gini}(\hat{\mathbf{u}}) = \frac{1}{d \sum_{p=1}^{d} x_p} \sum_{i=1}^{d} (2i - d - 1)\, x_i. \tag{24}$$

Higher Gini means updates are sparse in a few feature dimensions, while lower Gini indicates they are more uniformly spread across feature dimensions.

**Entropy ($\uparrow$).** Shannon entropy (Huang & Tran, 2019) of the $L_1$-normalized distribution $q[p] = \hat{\mathbf{u}}[p] / \sum_j \hat{\mathbf{u}}[j]$:

$$H(\hat{\mathbf{u}}) = \frac{-\sum_{p=1}^{d} q[p] \log q[p]}{\log d}. \tag{25}$$

Dividing by $\log d$ scales entropy to $[0, 1]$, where 0 denotes maximal sparsity with all updates in one feature dimension and 1 denotes maximal distribution with updates spread uniformly across dimensions. Entropy thus measures the diversity of feature dimensions updated during SFT.

Together, these metrics provide complementary views of sparsity: Top-$k$ Mass highlights whether updates are dominated by a small subset of feature dimensions, PR estimates the effective number of feature dimensions activated, Gini quantifies inequality across feature dimensions, and Entropy captures distributional diversity. We report values averaged across all Transformer blocks in the Table 4, reflecting parameter change patterns at the global level. Consistent improvements across all four metrics indicate that LoRA-B activates a broader and more stable set of parameters than LoRA-A.

Table 4: Sparsity metrics for LoRA-A and LoRA-B, averaged over 32 Transformer blocks. Results are reported for Query ($Q$), Key ($K$), and Value ($V$) projections, with Top-$k$ Mass computed at $k = 30\%$. LoRA-B consistently exhibits lower sparsity and greater distribution of updates compared to LoRA-A. **Bold** numbers indicate the better value per metric.

| Matrix | Query ($Q$) | | | | Key ($K$) | | | | Value ($V$) | | | |
|---|---|---|---|---|---|---|---|---|---|---|---|---|
| | Top-$k$ Mass ($\downarrow$) | PR ($\uparrow$) | Gini ($\downarrow$) | Entropy ($\uparrow$) | Top-$k$ Mass ($\downarrow$) | PR ($\uparrow$) | Gini ($\downarrow$) | Entropy ($\uparrow$) | Top-$k$ Mass ($\downarrow$) | PR ($\uparrow$) | Gini ($\downarrow$) | Entropy ($\uparrow$) |
| LoRA-A | 0.3492 | 3764.3338 | 0.0782 | 0.9979 | 0.3397 | 3839.3339 | 0.0645 | 0.9981 | 0.3542 | 3744.9923 | 0.0871 | 0.9982 |
| LoRA-B | **0.3250** | **4007.5958** | **0.0389** | **0.9997** | **0.3209** | **4027.5595** | **0.0330** | **0.9998** | **0.3431** | **3857.9129** | **0.0652** | **0.9992** |

Table 5: EDIT hyperparameter configuration for each benchmark and sequence length. Parameters were selected on validation sets to optimize the accuracy-efficiency trade-off.

| Dataset (Len) Parameter | Countdown | | | Sudoku | | | MATH500 | | | GSM8K | | | GPQA | | |
|---|---|---|---|---|---|---|---|---|---|---|---|---|---|---|---|
| | 128 | 256 | 512 | 128 | 256 | 512 | 128 | 256 | 512 | 128 | 256 | 512 | 128 | 256 | 512 |
| **All Blocks** | | | | | | | | | | | | | | | |
| Block Temperature ($\tau_{\text{blk}}$) | 1.0 | 1.0 | 1.0 | 1.0 | 1.0 | 1.0 | 1.0 | 1.0 | 1.0 | 1.0 | 1.0 | 1.0 | 1.0 | 1.0 | 1.0 |
| **First Block** | | | | | | | | | | | | | | | |
| Threshold ($\delta$) | 0.05 | 0.05 | 0.05 | 0.05 | 0.05 | 0.05 | 0.05 | 0.05 | 0.05 | 0.05 | 0.05 | 0.05 | 0.05 | 0.05 | 0.05 |
| Stability Span ($\Omega$) | 6 | 6 | 6 | 6 | 6 | 6 | 6 | 6 | 6 | 6 | 6 | 6 | 6 | 6 | 6 |
| **Subsequent Blocks** | | | | | | | | | | | | | | | |
| Threshold ($\delta$) | 0.05 | 0.55 | 0.45 | 0.1 | 0.45 | 0.05 | 0.1 | 0.05 | 0.025 | 0.05 | 0.025 | 0.025 | 0.05 | 0.05 | 0.05 |
| Stability Span ($\Omega$) | 6 | 12 | 12 | 6 | 12 | 6 | 6 | 6 | 6 | 6 | 6 | 6 | 6 | 6 | 8 |

## C   EXTENDED EXPERIMENTAL DETAILS

### C.1   COMPLETE HYPERPARAMETER CONFIGURATION

Table 5 provides the complete hyperparameter settings used in our experiments. These were selected using the validation protocol described in Section 3.2. We use a block length of 64 in EDIT.

### C.2   ADDITIONAL VISUALIZATIONS

Figure 6 provides additional evidence for the importance of preserving training-time metadata in LoRA-B. With LoRA rank $r = 128$ and output dimension 4096, the LoRA-B projection contains $4096 \times 128 = 524{,}288$ parameters. For visualization, we reshape this parameter tensor into a $256 \times 256 \times 8$ grid, where the Z-axis encodes the normalized AdamW update magnitudes (scaled to [0,255]). The visualization shows clear signatures, with strong update activity concentrated in specific regions, indicating parameters that receive consistent adjustments during fine-tuning and thus provide informative patterns for inference. In contrast, Figure 7 shows the LoRA-A projection with input dimension 4096 and rank 128 (also 524,288 parameters). LoRA-A displays much flatter patterns, suggesting only minimal parameter changes during the same training process.

### C.3   EARLY TERMINATION DIAGNOSTIC

Figure 8 shows the distribution of EDIT stopping steps for GSM8K at sequence length 512. Correct predictions typically stop later in the denoising process, while incorrect ones tend to halt earlier. This pattern aligns with GSM8K's longer reasoning chains—cases that require more structured computation generally stabilize at deeper denoising steps.

## D   THEORETICAL FOUNDATIONS OF EARLY TERMINATION IN EDIT

### D.1   SETUP AND NOTATION

At denoising step $t$, let $\mathcal{S}_t$ denote the visible tokens, and $\mathcal{I}_t = \mathcal{S}_{t-1} \cap \mathcal{S}_t$ the matched support between steps $t - 1$ and $t$. Let $\tilde{P}^{(t)}$ be the probability distribution on $\mathcal{I}_t$ obtained by restricting $P^{(t)}$

Figure 6: Visualization of LoRA-B parameter updates from training steps 105–120. A $4096 \times 128$ LoRA-B projection produces $524{,}288$ parameters, reshaped into a $256 \times 256 \times 8$ grid with the Z-axis showing normalized AdamW update magnitudes (scaled 0–255). The left diagram illustrates the indexing scheme, where each parameter is placed on the X–Y plane and its update magnitude is shown along the Z-axis, with the blue dot highlighting an example parameter and its upward link to the corresponding value. The right plot shows that a large subset of LoRA-B parameters receives frequent and strong updates, producing clear signatures of importance shaped by optimization dynamics. The animation is interactive and plays when clicked in Adobe Acrobat.

Figure 7: Visualization of LoRA-A parameter updates from training steps 105–120. A $128 \times 4096$ LoRA-A projection produces $524{,}288$ parameters, reshaped into a $256 \times 256 \times 8$ grid with the Z-axis showing normalized AdamW update magnitudes (scaled 0–255). The left diagram illustrates the indexing scheme, where each parameter is placed on the X–Y plane and its update magnitude is shown along the Z-axis, with the blue dot highlighting an example parameter and its upward link to the corresponding value. The right plot shows flatter patterns compared to LoRA-B, with only limited regions of noticeable activity, suggesting less distinctive signatures for guiding inference. The animation is interactive and plays when clicked in Adobe Acrobat.

to $\mathcal{I}_t$ and renormalizing. Define the per-step divergence

$$D_t = D_{\mathrm{KL}}(\tilde{P}^{(t)} \parallel \tilde{P}^{(t-1)}). \tag{26}$$

EDIT declares stability when $D_{t-\Omega+1}, \dots, D_t \leq \delta$ for some integers $\Omega \geq 1$ and threshold $\delta > 0$ (with fixed in-block temperature when forming $P^{(\cdot)}$).

We use the total variation distance $\mathrm{TV}(p, q) = \frac{1}{2} \sum_i |p_i - q_i|$ and Pinsker's inequality $\mathrm{TV}(p, q) \leq \sqrt{\frac{1}{2} D_{\mathrm{KL}}(p \parallel q)}$.

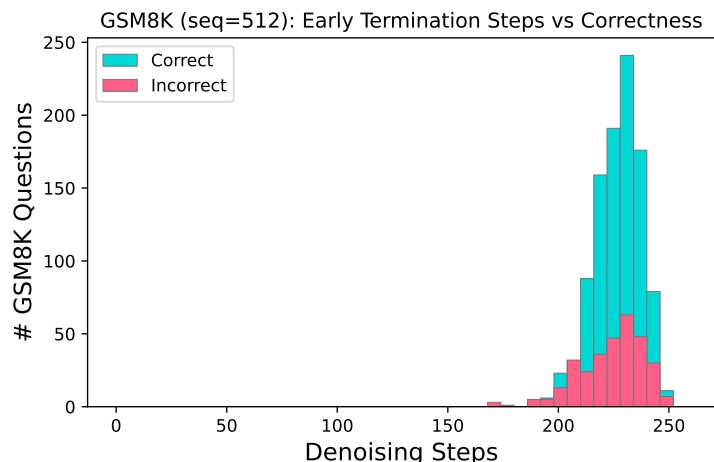

Figure 8: Distribution of EDIT termination steps on GSM8K (seq. 512). The x-axis shows denoising steps and the y-axis counts questions. Correct predictions tend to terminate at later steps.

## D.2 Multi-Step Control from Run-Length KL

**Lemma D.1** (Run-length KL implies multi-step TV bound). *If $D_{t-\Omega+1}, \ldots, D_t \leq \delta$, then*

$$TV(\tilde{P}^{(t)}, \tilde{P}^{(t-\Omega)}) \leq \sum_{r=t-\Omega+1}^{t} TV(\tilde{P}^{(r)}, \tilde{P}^{(r-1)}) \leq \Omega\sqrt{\frac{\delta}{2}}. \tag{27}$$

*Proof.* The result follows from the triangle inequality for total variation distance, then applying Pinsker's inequality to each summand. ☐

## D.3 Local Argmax Invariance at the Stopping Time

Let $i^*(t) = \arg\max_{s \in \mathcal{I}_t} \tilde{P}^{(t)}(s)$ and $m_t = \tilde{P}^{(t)}_{(1)} - \tilde{P}^{(t)}_{(2)}$ be the top-2 margin on $\mathcal{I}_t$.

**Theorem D.2** (Local argmax invariance certificate). *If $D_{t-\Omega+1}, \ldots, D_t \leq \delta$ and*

$$\Omega\sqrt{\frac{\delta}{2}} < \frac{1}{2}m_t, \tag{28}$$

*then $i^*(t') = i^*(t)$ for all $t' \in \{t - \Omega, \ldots, t\}$.*

*Proof.* If $i^*$ changed between $p = \tilde{P}^{(t)}$ and $q = \tilde{P}^{(t')}$, then $\text{TV}(p, q) \geq \frac{1}{2}(p_{(1)} - p_{(2)}) = \frac{1}{2}m_t$ (considering the mass that must move between the top two coordinates when the argmax changes). This contradicts Lemma D.1. ☐

**Interpretation:** When EDIT stops and the inequality holds, the predicted token on the matched support has been unchanged for the past $\Omega$ steps—providing a verifiable certificate attached to the stopping decision.

## D.4 Future-Step Robustness via Contraction

After the last unmasking in a block, let $K_r$ denote the Markov operator that maps $\tilde{P}^{(r-1)}$ to $\tilde{P}^{(r)}$ on the fixed support. Define the Dobrushin coefficient $\alpha(K_r) = \sup_{p \neq q} \frac{\text{TV}(pK_r, qK_r)}{\text{TV}(p,q)} \in [0, 1]$.

**Assumption D.3** (Local contraction post-unmasking). *There exists $\alpha < 1$ such that $\alpha(K_r) \leq \alpha$ for all $r \geq t + 1$ within the block. This property is standard for convergent Markov chains and can be verified empirically on validation data.*

**Theorem D.4** (Tail movement bound and global argmax preservation). *Under Assumption D.3, if $D_{t-\Omega+1}, \ldots, D_t \leq \delta$, then for any $s \geq 1$,*

$$TV(\tilde{P}^{(t+s)}, \tilde{P}^{(t)}) \leq \frac{\alpha^s}{1-\alpha} TV(\tilde{P}^{(t)}, \tilde{P}^{(t-1)}) \leq \frac{\alpha^s}{1-\alpha} \sqrt{\frac{\delta}{2}}, \tag{29}$$

*and thus $\sup_{s \geq 1} TV(\tilde{P}^{(t+s)}, \tilde{P}^{(t)}) \leq \frac{1}{1-\alpha} \sqrt{\frac{\delta}{2}}$.*

*If additionally*

$$\Omega \sqrt{\frac{\delta}{2}} + \frac{1}{1-\alpha} \sqrt{\frac{\delta}{2}} < \frac{1}{2} m_t, \tag{30}$$

*then $i^*(t+s) = i^*(t)$ for all $s \geq 0$ (argmax is preserved forever on the fixed support).*

*Proof.* One-step TV contracts by at most $\alpha$; summing the geometric tail yields the bound. The argmax preservation follows by the same margin argument as in Theorem D.2. □

### D.5 STABILITY OF LIPSCHITZ OBSERVABLES

**Theorem D.5** (Stability of Lipschitz functionals). *Let $F : \Delta \to \mathbb{R}$ satisfy $|F(p) - F(q)| \leq L \cdot TV(p, q)$ for all $p, q$. If $D_{t-\Omega+1}, \ldots, D_t \leq \delta$, then*

$$|F(\tilde{P}^{(t)}) - F(\tilde{P}^{(t-\Omega)})| \leq L \cdot \Omega \sqrt{\frac{\delta}{2}}. \tag{31}$$

*Under Assumption D.3,*

$$\sup_{s \geq 1} |F(\tilde{P}^{(t+s)}) - F(\tilde{P}^{(t)})| \leq \frac{L}{1-\alpha} \sqrt{\frac{\delta}{2}}. \tag{32}$$

*Proof.* Direct application of Lemma D.1 and Theorem D.4 with the Lipschitz property. □

### D.6 PRACTICAL CALIBRATION OF $(\delta, \Omega)$

Let $M$ denote the top-2 margin at EDIT's stopping time on a validation set, and $q_{1-\beta}$ its $(1 - \beta)$-quantile. Estimate a post-unmasking contraction bound by

$$\hat{\alpha} = \max_{\text{val instances, late } r} \frac{TV(\tilde{P}^{(r+1)}, \tilde{P}^{(r)})}{TV(\tilde{P}^{(r)}, \tilde{P}^{(r-1)})}. \tag{33}$$

**Corollary D.6** (PAC-style guarantee for the final answer). *Choose $(\delta, \Omega)$ to satisfy*

$$\Omega \sqrt{\frac{\delta}{2}} + \frac{1}{1-\hat{\alpha}} \sqrt{\frac{\delta}{2}} \leq \frac{1}{2} q_{1-\beta}. \tag{34}$$

*Then, with probability at least $1 - \beta$ over test instances, the top-1 token at EDIT's stopping time equals the top-1 token obtained by continuing denoising indefinitely (on the fixed support).*

*Proof.* Apply Theorem D.4 and the definition of $q_{1-\beta}$. □

**Reporting recommendation:** Alongside accuracy and step reductions, report the fraction of test instances that satisfy Corollary D.6 ("percentage of certified stops"). This quantifies how often EDIT halts with a provable correctness certificate.

### D.7 CERTIFIED EARLY TERMINATION

Theorem D.2 shows that when the aggregated run-length divergence is smaller than half of the top-2 margin, the predicted argmax token remains invariant across the past $\Omega$ steps. To assess how often this condition is met in practice, we evaluate the inequality at the stopping time on each benchmark with $\Omega = 2$, as shown in Table 6. An early stop is counted as certified if the condition of Theorem D.2 holds.

The condition is satisfied in about 86% of cases, indicating that the criterion of Theorem D.2 is frequently realized in practice.

Table 6: Certified early termination rates across benchmarks.

| Dataset | Countdown | GSM8K | MATH500 | Sudoku | Average |
|---|---|---|---|---|---|
| Certified (%) | 79.7 | 82.8 | 84.4 | 96.9 | 85.9 |

# E  TOKEN-WISE EDIT: PER-TOKEN FREEZING WITH CERTIFICATES

## E.1  LOCAL STABILITY STATISTICS AND RULE

Let $U \in \mathbb{R}^{d \times k}$ denote a fixed reasoning subspace (construction examples provided below). For each visible token $s$ at step $t$, define its subspace coordinates $g_s^{(t)} = U^\top f_s^{(t)} \in \mathbb{R}^k$ and the local distribution

$$Q_s^{(t)}(j) = \frac{\exp(|g_{s,j}^{(t)}|/\tau_{\text{sub}})}{\sum_{\ell=1}^k \exp(|g_{s,\ell}^{(t)}|/\tau_{\text{sub}})}, \quad j = 1, \ldots, k, \tag{35}$$

with fixed $\tau_{\text{sub}} > 0$.

Define the per-token KL $D_{s,t} = D_{\text{KL}}(Q_s^{(t)} \parallel Q_s^{(t-1)})$ and the run-length condition: token $s$ is locally stable at $t$ if $D_{s,t-r} \leq \delta_{\text{tok}}$ for $r = 0, \ldots, \Omega_{\text{tok}} - 1$. The token-wise EDIT freezes $s$ at $t$ (setting $f_s^{(t')} \equiv f_s^{(t)}$ for all $t' > t$) whenever this condition holds.

## E.2  PER-TOKEN CERTIFICATES

**Lemma E.1** (Local run-length bound). *If $D_{s,t-r} \leq \delta_{tok}$ for $r = 0, \ldots, \Omega_{tok} - 1$, then*

$$TV(Q_s^{(t)}, Q_s^{(t-\Omega_{tok})}) \leq \Omega_{tok} \sqrt{\frac{\delta_{tok}}{2}}. \tag{36}$$

*Proof.* Triangle inequality and Pinsker's inequality, as in Lemma D.1. $\square$

Let $j_s^*(t) = \arg\max_j Q_s^{(t)}(j)$ and $m_s(t) = Q_{s,(1)}^{(t)} - Q_{s,(2)}^{(t)}$ be the local top-2 margin.

**Theorem E.2** (Dominant subspace-component invariance per token). *If the condition of Lemma E.1 holds and $\Omega_{tok}\sqrt{\delta_{tok}/2} < \frac{1}{2}m_s(t)$, then $j_s^*(t') = j_s^*(t)$ for all $t' \in \{t - \Omega_{tok}, \ldots, t\}$.*

*Proof.* Identical to Theorem D.2 but applied to $Q_s^{(\cdot)}$. $\square$

## E.3  FREEZING SAFETY UNDER WEAK COUPLING

We quantify how freezing one token perturbs the global distribution $P^{(t)}$.

**Assumption E.3** (Weak cross-token coupling). *There exists $\beta_s \geq 0$ such that, if two states at step $r$ differ only in token $s$ by $\Delta_s$ (that is, $f_s^{(r)} \mapsto f_s^{(r)} + \Delta_s$), then their next-step global distributions satisfy*

$$TV(\tilde{P}^{(r+1)}, \tilde{P}'^{(r+1)}) \leq \beta_s \|\Delta_s\|_2. \tag{37}$$

*This $\beta_s$ can be estimated on validation by finite-difference probes.*

**Assumption E.4** (Post-unmasking contraction). *Within a block after the last unmasking, the Markov operators contract TV with coefficient $\alpha < 1$ as in Assumption D.3.*

**Theorem E.5** (Safety of freezing token $s$). *Suppose token $s$ satisfies the local stability condition at time $t$, and set*

$$\varepsilon_s = \max_{r \in \{t-\Omega_{tok}+1, \ldots, t\}} \|f_s^{(r)} - f_s^{(r-1)}\|_2. \tag{38}$$

*If token $s$ is frozen at $t$, then for all $u \geq 1$,*

$$TV(\tilde{P}_{frozen}^{(t+u)}, \tilde{P}_{unfrozen}^{(t+u)}) \leq \frac{\beta_s}{1-\alpha}\varepsilon_s. \tag{39}$$

*Consequently, if*

$$\Omega\sqrt{\frac{\delta}{2}} + \frac{\beta_s}{1-\alpha}\varepsilon_s < \frac{1}{2}m_t, \tag{40}$$

*where $m_t$ is the global top-2 margin at t, then the global argmax remains unchanged forever (on the fixed support) after freezing token s.*

*Proof.* One-step deviation is at most $\beta_s\varepsilon_s$ by Assumption E.3. Propagating under Assumption E.4 yields a geometric tail bound $\sum_{j\geq 0}\alpha^j\beta_s\varepsilon_s = \frac{\beta_s}{1-\alpha}\varepsilon_s$. Combine with Theorem D.2. $\square$

**Construction of $U$ and practical tuning:** A simple choice is to take $\bar{U}_B$ from Equation 3, compute its left singular vectors, and set $U$ to the top $k \in \{2,3,4\}$ vectors. Empirically, $k = 3$ or $k = 4$ provides good stability-efficiency trade-offs. On validation, choose $(\delta_{\text{tok}}, \Omega_{\text{tok}})$ to maximize frozen-token count subject to Theorem E.2's margin condition.

## F  SUBSPACE EDIT: REPLACING THE REASONING VECTOR BY A SUBSPACE

### F.1  DEFINITION

Let $U \in \mathbb{R}^{d\times k}$ with orthonormal columns ($k \geq 1$). Replace the scalar alignment in Equation 5 by a subspace score:

$$\text{Sim}_s^{(t)} = \|U^\top f_s^{(t)}\|_2 \quad \text{or} \quad \text{Sim}_s^{(t)} = \frac{\|U^\top f_s^{(t)}\|_2}{\|f_s^{(t)}\|_2} \quad \text{(subspace cosine)}, \tag{41}$$

and form $P^{(t)}$ from Equation 6 with the same fixed in-block temperature $\tau_{\text{blk}}$. All other components of EDIT (matched-support renormalization, KL divergence, run-length rule) remain unchanged.

### F.2  INHERITED GUARANTEES

**Proposition F.1** (Guarantees are shape-agnostic in the similarity). *The statements and proofs of Lemma D.1, Theorems D.2–D.5, and Corollary D.6 hold verbatim under the subspace similarity above.*

*Proof.* The guarantees depend only on the distributions $\tilde{P}^{(\cdot)}$ and their KL/TV relations. The construction of $\text{Sim}_s^{(t)}$ enters only through $P^{(t)}$'s definition with a fixed temperature. $\square$

**Constructing $U$:** Two practical choices are available. First, perform SVD of $\bar{U}_B$ (Equation 3) and retain the top $k$ left singular vectors. Second, use CCA between step-averaged preconditioned gradients and activations $\{f_s\}$ in the chosen module. Ablations suggest small $k$ values (2–4) are sufficient and can stabilize earlier than single-vector approaches, potentially offering improved efficiency-accuracy trade-offs.

## G  GRADIENT-BASED JUSTIFICATION FOR EARLY TERMINATION

To determine when denoising steps can be truncated safely, we compare inference pseudo-gradients with the SFT gradients $G_{k,B}$ on LoRA-B layers. During inference, at each denoising step $t \in \{1, \ldots, T_b\}$, the model produces logits $z_t(s)$ for every token position $s$ in the block of length $L$. Since no ground-truth labels are available at inference time, the only informative signal comes from the evolution of predictive distributions across steps. We denote $p_\theta(z_t(s))$ as the model's prediction for token $s$ at step $t$ and $p_\theta(z_{t+1}(s))$ as the refined prediction at step $t+1$. The KL divergence between them measures how much the model's belief changes across steps, with larger values indicating ongoing refinement and smaller values indicating stabilization. We therefore define the pseudo-gradient as

$$\tilde{G}_{t,B} = \nabla_B \sum_{s\in S_{t+1}} \text{KL}\left(p_\theta(z_t(s)) \,\middle|\, p_\theta(z_{t+1}(s))\right), \tag{42}$$

where $S_{t+1}$ denotes the visible token set at step $t+1$. We compute the pseudo-gradient by evaluating the KL divergence between consecutive predictive distributions at steps $t$ and $t+1$, restricted to $S_{t+1}$ so that only effective (unmasked) tokens contribute. Backpropagating this divergence through the LoRA-B parameters yields $\tilde{G}_{t,B}$, and we record its root-mean-square (RMS) magnitude as a scalar summary for step $t$. Repeating this across all denoising steps produces a trajectory of pseudo-gradients that characterizes the sensitivity of inference dynamics.

For training, we take the gradients $G_{k,B}$ observed at each SFT step $k$, compute their RMS magnitudes, and summarize them by a mean $\mu_{\text{SFT}}$ and a variance band. These gradients fluctuate around the mean within a bounded band, defining the stable regime in which the model was optimized. By overlaying the inference pseudo-gradients $\tilde{G}_{t,B}$ with this SFT reference, we obtain a principled test of alignment. Convergence is declared when pseudo-gradients (1) approach $\mu_{\text{SFT}}$ and (2) remain within this band, beyond which further denoising adds cost without benefit. Initially, $\tilde{G}_{t,B}$ deviates from the SFT regime, but after several iterations it reaches a convergence point $t_{\text{conv}} = \arg\min_t |\text{RMS}(\tilde{G}_{t,B}) - \mu_{\text{SFT}}|$, after which the pseudo-gradients oscillate around the SFT mean $\mu_{\text{SFT}}$ in a manner statistically consistent with $G_{k,B}$. This indicates that inference has entered the same training-consistent regime, and further denoising steps add computation without providing additional alignment benefit.

Empirically, on the GPQA benchmark (sequence length 128) we analyze the second diffusion block and observe in Figure 9 that the pseudo-gradients reach a convergence point (marked by the yellow ▼) at the 19-th denoising step. Beyond this point, they fluctuate stably around the SFT mean, indicating entry into the training-consistent regime. Terminating at ∼20 steps per block therefore preserves fidelity while reducing computation, consistent with Table 2, which shows an average of 40.3 steps for two diffusion blocks (*i.e.*, ∼20 steps each) on GPQA, and with Table 1, which confirms that accuracy remains competitive.

After the convergence point at step 19, the pseudo-gradients stay within the SFT variance band, oscillating around the mean, but a spike appears near step 25. This behavior may reflect the dynamics of late denoising. At this stage, most unmasked (visible) tokens have stabilized, while updates focus on the remaining masked positions. These late updates often involve low-information elements such as function words, punctuation, or minor phrasing, which carry little semantic weight but can still trigger abrupt shifts in the predictive distribution. Importantly, because these refinements concern filler-like tokens, they do not compromise the earlier convergence, which already ensures fidelity comparable to full denoising.

We also show additional examples in Figure 10, which presents Countdown, Sudoku, MATH500, and GSM8K. Their convergence points occur at the 20-th, 18–19-th, 19-th, and 23-rd steps respectively. In each task, terminating around these points preserves fidelity while reducing computation, consistent with the average step counts reported in Table 2 (40.4, 38.3, 38.1, and 42.8 steps for two blocks). These results demonstrate that pseudo-gradient convergence provides a consistent signal of reasoning step completion across diverse benchmarks.

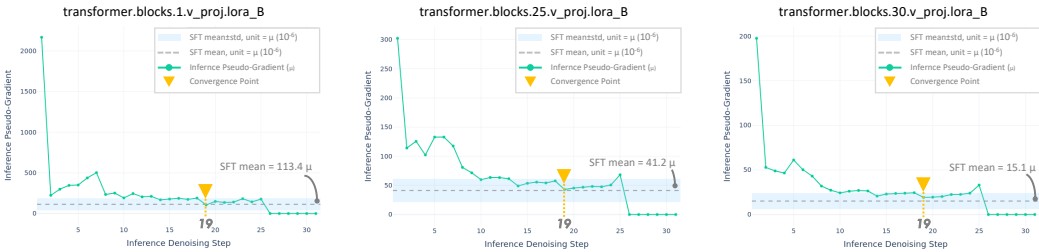

Figure 9: Gradient-based analysis of training–inference alignment on GPQA (sequence length 128, second diffusion block). The curve shows the RMS pseudo-gradients $\tilde{G}_{t,B}$ across denoising steps, compared against the SFT gradient mean (dashed) and variance band (shaded). The convergence point (yellow ▼) occurs at the 19-th step, after which pseudo-gradients oscillate stably around the SFT mean. This alignment indicates that terminating at ∼20 steps per block maintains fidelity comparable to full denoising while reducing computation, consistent with Table 2 (40.3 steps for two blocks).

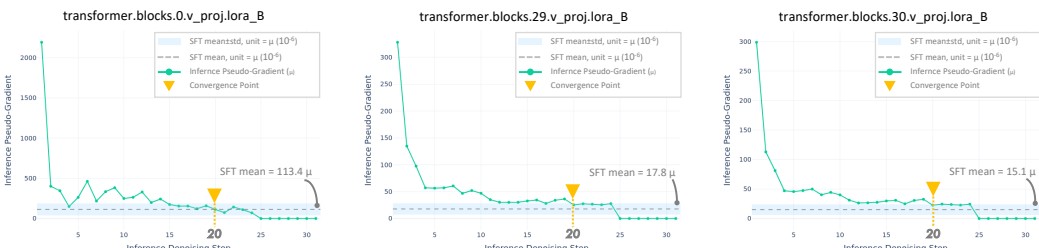

(a) Countdown (sequence length 128, second diffusion block). The convergence point (yellow ▼) occurs at the 20-th step. Terminating at ∼20 steps per block maintains fidelity comparable to full denoising while reducing computation, consistent with Table 2 (40.4 steps for two blocks).

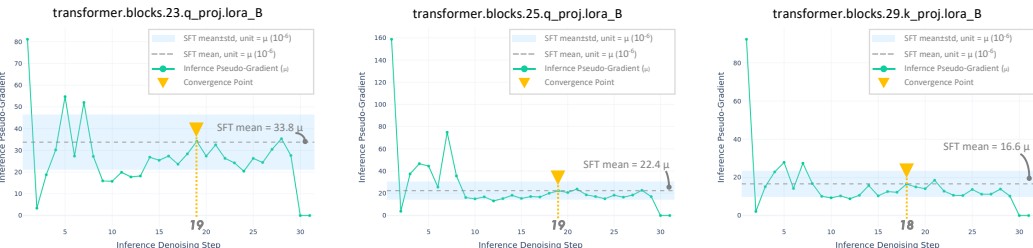

(b) Sudoku (sequence length 128, second diffusion block). The convergence point (yellow ▼) occurs at the 18-th and 19-th steps. Terminating at ∼19 steps per block maintains fidelity comparable to full denoising while reducing computation, consistent with Table 2 (38.3 steps for two blocks).

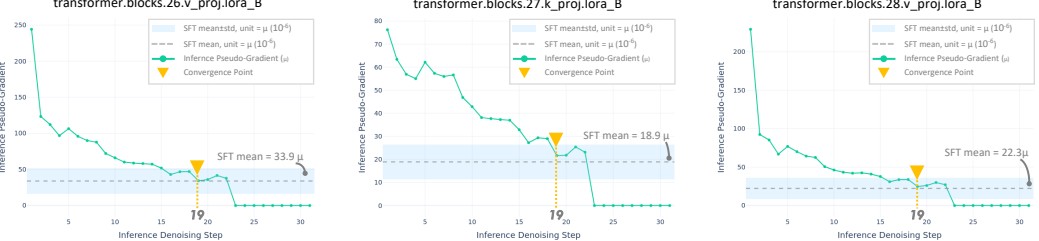

(c) MATH500 (sequence length 128, second diffusion block). The convergence point (yellow ▼) occurs at the 19-th step. Terminating at ∼19 steps per block maintains fidelity comparable to full denoising while reducing computation, consistent with Table 2 (38.1 steps for two blocks).

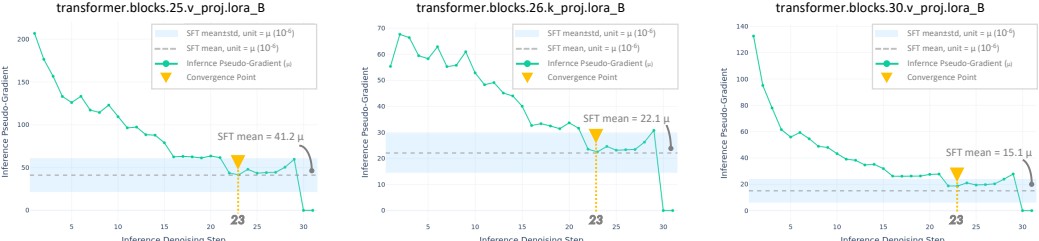

(d) GSM8K (sequence length 128, second diffusion block). The convergence point (yellow ▼) occurs at the 23-rd step. Terminating at ∼22 steps per block maintains fidelity comparable to full denoising while reducing computation, consistent with Table 2 (42.8 steps for two blocks).

Figure 10: Gradient-based analysis of training–inference alignment across four benchmarks (sequence length 128, second diffusion block). Each plot shows RMS pseudo-gradients $\tilde{G}_{t,B}$ across denoising steps, compared against the SFT gradient mean (dashed) and variance band (shaded). The yellow ▼ marks the convergence point, after which pseudo-gradients oscillate around the SFT mean. These examples illustrate that across diverse reasoning tasks (Countdown, Sudoku, MATH500, GSM8K), pseudo-gradients consistently stabilize near the convergence step. Termination at or near this point maintains fidelity comparable to full denoising while reducing computation, consistent with Table 2.

