# OpenReview forum: "EDIT: Early Diffusion Inference Termination for dLLMs Based on Dynamics of Training Gradients"
_ICLR.cc/2026/Conference — Submitted to ICLR 2026_

### Official Review · Reviewer_uwNg · 2025-10-26

**Soundness:** 1
**Presentation:** 2
**Contribution:** 1
**Rating:** 2
**Confidence:** 4

**Summary:**

This paper provides a mechanism to early-stop the diffusion process in a dLLM.  Basically, we collect information during LoRA fine-tuning about which dimensions of the query output projection (in the last layer of the Transformer) get the most consistent gradients, as per updates to the LoRA B matrix.  Then, during inference, we look at the dimensions of unmasked tokens in the representations after the same query projection, and see if they are aligning with the FT gradients.  If they are, we say that "reasoning stability" has been achieved, and we stop the diffusion process.  Some experiments assess whether this can really save diffusion steps on downstream tasks.

**Strengths:**

It's well-motivated to try to make use of the gradient information that was collected during training (or fine-tuning) and re-use that during inference.  Diffusion seems like a great place to start.

I think even the specific idea to use gradient momentum info from reasoning-based SFT to see if generation has arrived at a "reasoning"-like solution is not necessarily a bad idea.

Studying diffusion LLMs are also a very wide-open space so testing dLLMs like LLaDA-8B on downstream tasks, and finding out what the weak points are, is good to do.

Good job to have a discussion of limitations (in the conclusion), however brief.

**Weaknesses:**

Overall, the approach here just feels very premature, like, as a practitioner, I am really unsure whether simpler things would work, or whether other choices could unlock much bigger gains.  There's not a lot of theory or empirical rigour behind the findings, e.g., what if we used an Oracle and stopped at the optimal number of steps in each case, what are the maximum speedups that we could get, or the best improvements in quality?  It's just not well-scoped in the current submission.  This lack of rigor and testing of alternative ideas, plus a lack of comparison to baselines, and some questionable experimental decisions, makes me think this paper is not yet valuable to the community.

From my perspective, the scope of other things that COULD be tried is very large, and it's not clear how important the choices made in the paper actually are:
- What about just using the parameters themselves, or momentum itself, or just the evolution tensor, rather than the evolution tensor after the reduction?
- Why just the Query projection?  Why just the last layer?
- Why Cosine versus other measures?  Why KL divergence versus other measures?
- What if we didn’t ensure stability with Ω consecutive scores below δ, but did something else?

The current method introduces new task-specific (!) hyperparameters:
- δ, Ω, τblk
- Fundamentally, I don’t think we should be tuning hyperparameters on specific downstream tasks, as this confounds model comparison – how can we compare to models that weren’t fine-tuned, e.g., on the “Countdown” task?  Update: actually, this paper does make this mistake.
- Did we tune the number of diffusion steps on each task as well for the baselines?  It seems like you just compare at different max denoising steps (set depending on sequence length), but the EDIT count is below the smallest that you tested, so it makes me wonder if fewer steps would be better for these other ones.

Soundness:
- The fact we didn’t compare to any other methods of early stopping for denoising… this really surprised me.  Like, accepting and freezing unmasked tokens when their probability crosses a threshold?  You mentioned “output stability or confidence” and "entropy" so why not test those?  What's the downside?
- I mean, the fact we get different (and often better) accuracy numbers is surprising to me, and seems to reflect some kind of bias in the trained models that can be alleviated through hyperparameter tuning, which was only done with EDIT.  Fundamentally, I would not expect more steps to impair a diffusion model, although I believe there is prior work showing accuracy does not increase monotonically in number of steps.  Perhaps you should have plotted accuracy versus steps for the baselines for Countdown and then if EDIT is below this curve, that would alleviate some of my concerns.

Nitpicks:
- I think since the fundamental idea of this paper is that SFT reveals reasoning patterns in the activations, you should really explain the SFT dataset in more detail, right?
- Oh man, Tables 1 and 2, those are quite tiny fonts!  Is there no limit to how small we’d make them???
- Prior work specifically using momentum to identify important params: Dettmers - Sparse Networks from Scratch - Faster Training without Losing Performance - 1907.04840v2
- You know, diffusion itself does allow “guidance” in the form of gradients to be applied during the generative process.  I think maybe you could link your pseudo-gradients to this theory a bit better.

**Questions:**

- When we convert alignment scores into a probability distribution, what is it a distribution over?   There are many probabilistic arguments made about this distribution, but I don’t really understand, e.g., what domain the “support” of this distribution lives in.
- Is full-parameter SFT or CPT or even RL a limitation of your method, or just a limitation of your experimental evaluation?  What about other optimizers, e.g., Muon?

---

> ### Author Response · Authors · 2025-12-03
> **Summary: Momentum, Cosine/KL choice, Stability rule, Threshold, Comparison, Optimizer-agnostic design**
>
> # Using momentum as the stopping signal
> Thank you for the suggestion. AdamW’s update rule combines both momentum and variance information, and EDIT relies on this combined update. We also evaluated using momentum alone as the stopping signal. It gives a reasonable trend, but the stopping point varies more and the accuracy is slightly lower than with EDIT (results in the table below).
> | Method                  | Countdown@128 | Countdown@256 | Countdown@512 |
> |-------------------------|----------------|----------------|----------------|
> | EDIT (momentum only)    | 23.4           | 29.7           | 25.4           |
> | EDIT (combined update)  | **28.9**       | **31.6**       | **27.7**       |
>
> These results indicate that momentum is informative, but the combined update signal offers a more stable basis for determining when denoising has settled.
>
> # Design choices and architectural components
> Thank you for raising this concern. The reference to the Query projection in the final layer in Section 2.2.1 is meant to illustrate the behavior shown in Figure 4, not to imply that EDIT relies exclusively on that component. In practice, EDIT evaluates LoRA-B activations from the Q/K/V projections across layers and uses their step-to-step stability as the stopping signal. This provides a consistent way to connect the update directions learned during SFT with the model’s denoising trajectory.
>
> We agree that other formulations—such as isolating specific projections or choosing different layers—could also be explored. Our goal in this work is to examine whether training-derived update information can support an effective halting signal, and the formulation we adopt is sufficient to study this question across several reasoning benchmarks.
>
> # Choice of cosine similarity and KL divergence
> We use cosine similarity to track directional alignment and KL divergence to assess distributional change. Cosine is scale-invariant, which helps when activation magnitudes vary across denoising steps, and KL provides a straightforward measure of how distributions shift over time.
>
> Other measures could also be used, but our goal is to test whether training-derived update information can provide a stable stopping signal. Cosine and KL give a clean and consistent criterion for this purpose across the benchmarks we study.
>
> # Choice of the $\Omega$-step stability rule
> EDIT uses an $\Omega$-step window to avoid reacting to brief fluctuations that appear in the very early denoising steps. A short consistency check makes the stopping point more stable while keeping the rule simple and lightweight. In practice, this criterion works reliably across the benchmarks we evaluate.
>
> # Task-specific thresholds and comparison to baselines
> We tune $\delta$ and $\Omega$ once per task using a small validation split, and we agree this introduces some tuning overhead. In practice, however, a narrow range $\delta \approx$0.05-0.1, $\Omega \approx$6-12 works reliably across benchmarks, so EDIT does not depend on fine-grained, task-specific adjustments.
>
> Regarding comparison to baselines, we use the standard diffusion schedules defined by each sequence length. We do not retune the number of denoising steps per task for the baselines; instead, we follow the default settings so that all methods share the same computation budget. EDIT uses a data-driven stopping rule, so its step count can fall below this schedule, but the baselines are evaluated under their default configurations.
>
> # Comparison to early-stopping baselines
> EDIT uses training-time optimizer metadata together with Q/K/V activation patterns to determine when the denoising process has stabilized. This gives a stopping rule based on how internal activations change during denoising, using the training-aligned update information rather than relying on output-level heuristics.
>
> # What is the distribution over, and what is its support?
> The probability distribution is defined over the visible (unmasked) tokens at the current denoising step. At step $t$, we compute an alignment score for each visible token in the set $S_t$ and apply a softmax over only those tokens. In other words, the distribution’s support is the visible-token set $S_t$ at that step.
>
> When comparing step $t−1$ and step $t$, the visible sets differ. We therefore restrict both distributions to the intersection $I_t = S_{t-1} ∩ S_t$, renormalize on this shared support, and compute KL divergence on $I_t$. All probabilistic arguments in the paper are based on this matched-support domain.
>
> # Full-parameter SFT/CPT/RL and Other optimizers
> Our use of LoRA-based SFT is an experimental choice—it simplifies extracting update dynamics and keeps metadata small. EDIT is optimizer-agnostic and only needs aggregated update statistics; it also works with Lion optimzer. Full-parameter SFT, CPT, or RL are compatible—they increase metadata size but do not change EDIT’s core mechanism of comparing training-time update patterns to inference-time activations.

---

### Official Review · Reviewer_3GSH · 2025-10-28

**Soundness:** 2
**Presentation:** 2
**Contribution:** 2
**Rating:** 4
**Confidence:** 3

**Summary:**

The paper proposes **EDIT (Early Diffusion Inference Termination)**, an adaptive early inference termination rule for diffusion LLMs (dLLMs). During SFT with LoRA on Q/K/V, the method aggregates AdamW update statistics into a compact, feature-aligned “AdamW evolution” vector ($u$) that encodes which parameters consistently carried the learning signal. At inference, EDIT computes per-step cosine similarities between token activations and $u$, and halts denoising for tokens once a stability condition is met. The authors provide extensive experiments supporting the claim that training dynamics can guide safe early termination.

**Strengths:**

* **Insightful diagnostics.** Gradient-based “pseudo-gradient” alignment with SFT gradients; domain-wise breakdowns (e.g., GPQA subdomains) and LoRA-A vs. LoRA-B sparsity analyses justify design choices.
* **Practical gains with tiny overhead.** Reported step reductions up to ~68% and a ~1.5 MB metadata footprint for a 32-block model are compelling for deployment.

**Weaknesses:**

* **Metadata extraction.** EDIT relies on training-time metadata extraction. Many released checkpoints does not expose training recipe, does the selection of training recipe (dataset, hyperparam) affect extraction.
* **Scope of validation.** Experiments center on a single dLLM family (LLaDA-8B) and five reasoning tasks on one hardware stack. It’s hard to assess robustness across models, sizes, datasets.
* **Task-tuned thresholds.** ($\delta,\Omega$) are selected per task via validation for an accuracy/steps trade-off. This introduces tuning burden and potential brittleness under distribution shift.
* **Poor presentation** Figures and tables are too small comparing to captions, such as Figure 3/4/5, Table 1/2. Certain figures in the paper have strange frames, such as Figure 2/4/8. The theory seems to be ad-hoc.

**Questions:**

* **Baselines & fairness.** How does EDIT compare to strong adaptive early inference termination rule for dLLMs under identical compute budgets—and to autoregressive early-exit baselines on similar tasks? **The paper only compare EDIT with plain baseline, However, a lot of dLLM acceleration methods already exists**, such as [1][2].
* **Unclear Intuition** What is the intuition of $u$? what makes similarity between each token’s activation and the AdamW evolution vector important? The paper does not provide a clear intuition.
* **Metadata extraction.** Does SFT training recipe affects the performance of EDIT? How does performance change under domain shift from the SFT distribution?
* **Adaptive calibration.** Can $(\delta,\Omega)$ be set online *without* task-level validation?

[1] Wu C, Zhang H, Xue S, Liu Z, Diao S, Zhu L, Luo P, Han S, Xie E. Fast-dllm: Training-free acceleration of diffusion llm by enabling kv cache and parallel decoding. arXiv:2505.22618. \
[2] Li P, Zhou Y, Muhtar D, Yin L, Yan S, Shen L, Liang Y, Vosoughi S, Liu S. Diffusion language models know the answer before decoding. arXiv:2508.19982. \
[3] Ben-Hamu H, Gat I, Severo D, Nolte N, Karrer B. Accelerated Sampling from Masked Diffusion Models via Entropy Bounded Unmasking, NeurIPS 2025.

---

> ### Author Response · Authors · 2025-12-03
> **Summary: Metadata, Domain shift, Validation, Tuning, and Baselines**
>
> We appreciate the reviewer’s thoughtful feedback and respond to the points below.
>
> # Metadata extraction
> We acknowledge that many released checkpoints do not preserve optimizer states, which limits where EDIT can be applied. Our method requires access only to these optimizer moments, and we highlight that this metadata provides useful signals about the model’s learned update dynamics. When available, it allows EDIT to relate training-time behavior to inference-time stability and improve efficiency without altering the base model.
>
> # Domain shift
> EDIT relies on the optimizer moments learned during SFT, while the stopping criterion is determined at inference based on when the model’s internal updates stabilize. This signal does not require the test distribution to mirror the SFT data. In our evaluation, tasks with different reasoning characteristics—including arithmetic, logic, and symbolic manipulation—show stable patterns of convergence, and EDIT halts reliably across all of them.
>
> # Scope of validation
> EDIT is lightweight and does not depend on a specific architecture—it only uses the optimizer-derived update information and the model’s inference trajectory—so the mechanism is applicable beyond this particular model. The five benchmarks cover different forms of reasoning, providing varied conditions for examining EDIT’s behavior. Broadening the evaluation to additional model sizes and architectures would offer further context and can be pursued in follow-up work.
>
> # Task-tuned thresholds and Adaptive calibration
> EDIT selects ($\delta$, $\Omega$) once per task using a small validation split. We agree that this introduces some tuning overhead. In practice, values in the range $\delta \approx$0.05–0.1 and $\Omega \approx$6–12 work reliably across several tasks, indicating that EDIT does not depend on fine-grained tuning. Although EDIT does not adjust these thresholds online, the stopping signal is computed during inference and does not rely on task-level labels. As a result, EDIT can run with default settings when task-specific validation is not available.
>
> # Baselines & fairness
> We appreciate the reviewer’s suggestion. In addition to the SFT baseline reported in the paper, we include comparisons with two recent dLLM acceleration methods, Fast-dLLM [1] and Prophet [2], under the same 0-shot evaluation setting. The results are shown in the tables below. The symbol ($\dagger$) denotes results reproduced from the Fast-dLLM open-source implementation. We also note that EDIT reaches its GSM8K@256 performance using $\approx$ 103.5 steps, while Prophet requires $\approx$160 steps.
>
> | Method                            | GSM8K@256 (0-shot) | GPQA@256 (0-shot) |
> | --------------------------------- | ------------------ | ----------------- |
> | EDIT (Ours)                   | 77.6          | **27.7**          |
> | LLaDA (SFT) + Cache (from [1]) †   | 72.0               | 22.1              |
> | LLaDA (SFT) + Parallel (from [1]) †| 73.1               | 26.3              |
> | Fast-dLLM [1]†                     | 71.6               | 22.5              |
> | Prophet [2]                       | **77.9**           | 25.7              |
>
> # Intuition behind u and the similarity measure
> The vector $\mathbf{u}$ is derived from the optimizer moments accumulated during SFT and reflects the update direction that the model repeatedly moved toward at a given projection layer. Intuitively, $\mathbf{u}$ summarizes how the model learned to adjust its computed representations during training.
> During inference, we measure how each token’s activation aligns with $\mathbf{u}$. When activations continue to change relative to $\mathbf{u}$, the model is still adjusting its internal representation in a way that resembles ongoing “reasoning.” Once this alignment stabilizes, the model’s updates stop moving in directions emphasized during training, indicating that the representation has settled. EDIT uses this stability as the signal that further denoising steps are no longer contributing meaningfully.
>
> # Presentation
> Thank you for pointing this out. We have enlarged Tables 1 and 2 and increased the font sizes in Figures 3, 4, and 5 to improve readability. We also removed the unintended frames in Figures 2, 4, 9, and 10.

---

### Official Review · Reviewer_GDvF · 2025-11-01

**Soundness:** 3
**Presentation:** 3
**Contribution:** 3
**Rating:** 6
**Confidence:** 3

**Summary:**

The paper introduces EDIT, an inference-time early termination rule for diffusion LLMs that reuses training-time optimizer information. During SFT with LoRA, the method aggregates AdamW moments into a compact “AdamW evolution” vector $u$ that encodes parameter-importance patterns; at inference it measures cosine alignment between token activations and $u$, converts these to per-step distributions on visible tokens, and stops when matched-support KL divergence stays below a threshold for $\\Omega$ consecutive steps. The theory shows a multi-step control bound $TV \\le \\Omega\\sqrt{\\delta/2}$ and a margin condition that preserves the argmax, yielding PAC-style certificates for chosen $(\\delta,\\Omega)$. Empirically, on LLaDA-8B across Countdown, Sudoku, MATH500, GSM8K, and GPQA, EDIT reduces denoising steps by 11.8% to 68.3% with comparable accuracy on most tasks; storage overhead is about 0.02% of an 8 GB model.

**Strengths:**

* Clear definition of the AdamW evolution signal and why LoRA-B is preferred, with sparsity metrics and visualizations.
* Matched-support KL and multi-step TV bounds yield simple certificates and a PAC-style calibration rule.
* Minimal storage and implementation overhead, with complexity lower than attention and integration as a wrapper at inference.
* Certified early stop rates reported across tasks, indicating practical realizations of the theory.

**Weaknesses:**

* Hyperparameter selection uses per-task validation sets and a grid over $(\\delta,\\Omega)$; robustness to mis-tuning or cross-task portability is not thoroughly analyzed.
* GSM8K at length 512 shows a noticeable accuracy drop with EDIT, which deserves a short diagnostic beyond the brief discussion. Minor.

**Questions:**

* Please report end-to-end system metrics for one representative task and length (e.g., Countdown@256): wall-clock per instance, tokens per second, and peak GPU memory, alongside the step reductions already shown in Table 2. This is a light logging change.
* For GSM8K at length 512, please add a short diagnostic: either (i) a histogram of early-stop step vs. correctness, or (ii) a 2×2 grid over $(\\delta,\\Omega)\\in\\{0.05,0.1\\}\\times\\{6,12\\}$ reporting accuracy and mean denoising steps. One figure or a small table is sufficient.

---

> ### Author Response · Authors · 2025-12-03
> **Summary: Efficiency, Diagnostics, and Hyperparameter**
>
> We appreciate the reviewer’s helpful suggestion and include the analysis below.
>
> # End-to-end system metrics (Countdown@256)
> | Method   | Wall time per instance | Tokens per second | Peak GPU memory (GB) | Step reduction (%) |
> | -------- | ---------------------- | ----------------- | -------------------- | ------------------ |
> | Baseline | 11.3                   | 22.6              | 17.1                 | –                  |
> | EDIT | **5.7**                | **44.4**          | 17.4                 | **68.3**           |
>
> # GSM8K@512 Diagnostic
> We show in Figure 8 (Section C.3) a histogram summarizing early-stop steps for GSM8K at sequence length 512. Correct predictions tend to stop at later denoising steps, while incorrect predictions more often halt slightly earlier. This pattern is consistent with GSM8K requiring longer reasoning chains.
>
> # Hyperparameter selection
> EDIT selects ($\delta$, $\Omega$) using a small per-task validation split. In our experiments, a narrow range of values ($\delta$ $\approx$ 0.05–0.1, $\Omega$ $\approx$ 6–12) produces consistent behavior across multiple tasks, suggesting reasonable tolerance to moderate mis-tuning.

---

### Official Review · Reviewer_KgfN · 2025-11-01

**Soundness:** 3
**Presentation:** 3
**Contribution:** 3
**Rating:** 6
**Confidence:** 3

**Summary:**

This paper proposes EDIT to speed up diffusion-style large language models. The key idea is to reuse training-time optimizer dynamics that are normally discarded: aggregated AdamW statistics on LoRA-B are compressed into a compact per-block “pathway vector” u. During inference, EDIT measures how well current visible-token activations align with u, constructs a distribution over the intersection of visible tokens across consecutive steps, and tracks a matched-support KL divergence. If the KL stays below a threshold δ for Ω consecutive steps, the method early-stops the diffusion process.
It provides theoretical support by bounding total-variation distance from KL and using a margin condition to argue that, with suitable (δ, Ω), early stopping will not change the final prediction. Empirically, EDIT reduces denoising steps by 11.8–68.3% on multiple reasoning benchmarks with little to no loss in accuracy. The approach requires only lightweight metadata extracted at training time, no architectural changes at inference, and minimal runtime overhead.

**Strengths:**

1.The method is clear, novel and well-supported by theory.

2.Without changing the model's reasoning structure, it enables the addition during training and no modification during inference, which differentiates it from previous efficiency-enhancing methods that require modifying the decoding or architecture.

3.The paper is easy to read, motivations are well connected.

**Weaknesses:**

1.The reliance on training metadata requires access to the optimization trajectory during the training phase, which is insufficient for scenarios involving closed-source models or those with only final checkpoint files.

2.Validation focuses on LoRA-SFT, it remains unclear how EDIT performs with full-parameter finetuning, other adapters, or different optimizers

3.Small accuracy dips appear in longer sequences, and the paper does not deeply analyze when EDIT stabilizes too early.

**Questions:**

1.Have you tried this method on other optimizers and adapters?

2.Under the influence of prompt form changes, noise disturbances, or adversarial interventions, is it more likely to prematurely stop? Have you ever encountered the situation of prematurely stopping?

---

> ### Author Response · Authors · 2025-12-03
> **Summary: Metadata requirement, Generality tests, and Stability behavior**
>
> We thank the reviewer for the thoughtful comments and constructive feedback. We address the points as follows.
>
> # Requirement of training metadata
> We agree that EDIT depends on the availability of optimizer moment statistics from fine-tuning. This is indeed a limitation: if the optimizer state is not preserved, EDIT cannot be applied. At the same time, these statistics are automatically produced in standard SFT workflows and are inexpensive to store, so EDIT is naturally suited to settings where fine-tuning is performed or where metadata can be retained. We acknowledge this limitation in the Conclusion and Future Directions section, and note that training metadata can support more holistic and efficient inference.
>
> # Applicability beyond LoRA-SFT, and use with other optimizers/adapters
> EDIT relies only on optimizer moments and projection-layer activations, not LoRA-specific structure. To assess compatibility across training setups, we applied EDIT to two variants: (1) switching the optimizer from AdamW to Lion, and (2) switching the adapter from LoRA to DoRA. The Countdown results are shown below.
>
> ### (Accuracy) Lion Optimizer + LoRA Adapter
> | Method      | Optimizer | Adapter | Countdown@128 | Countdown@256 | Countdown@512 |
> | ----------- | --------- | ------- | ------------- | ------------- | ------------- |
> | LLaDA (SFT) | Lion      | LoRA    | 33.6          | 21.5          | **30.1**      |
> | EDIT    | Lion      | LoRA    | **34.4**      | **26.6**      | 28.5          |
>
> ### (Steps) Lion Optimizer + LoRA Adapter
> | Method      | Optimizer | Adapter | Countdown@128 | Countdown@256 | Countdown@512 |
> | ----------- | --------- | ------- | ------------- | ------------- | ------------- |
> | LLaDA (SFT) | Lion      | LoRA    | 64            | 128           | 256           |
> | EDIT   | Lion      | LoRA    | 33.6          | 30.2          | 192.3         |
>
> ### (Accuracy) AdamW Optimizer + DoRA Adapter
> | Method      | Optimizer | Adapter | Countdown@128 | Countdown@256 | Countdown@512 |
> | ----------- | --------- | ------- | ------------- | ------------- | ------------- |
> | LLaDA (SFT) | AdamW     | DoRA    | **36.7**      | 27.3          | **25.4**      |
> | EDIT    | AdamW     | DoRA    | 28.5          | **29.3**      | 24.2          |
>
> ### (Steps) AdamW Optimizer + DoRA Adapter
> | Method      | Optimizer | Adapter | Countdown@128 | Countdown@256 | Countdown@512 |
> | ----------- | --------- | ------- | ------------- | ------------- | ------------- |
> | LLaDA (SFT) | AdamW     | DoRA    | 64            | 128           | 256           |
> | **EDIT**    | AdamW     | DoRA    | 44.3          | 40.6          | 135.1         |
>
> # Accuracy dips on longer sequences
>
> The small accuracy decreases observed on some long-sequence settings reflect differences in how much useful reasoning occurs during later denoising steps. Certain tasks (e.g., Countdown) stabilize earlier in the denoising process, while others (e.g., GSM8K) continue to benefit from additional refinement later on. EDIT halts once the alignment statistics stop changing across consecutive steps, and tasks reach this stability at different points in the denoising process.
> As a result, the accuracy variation aligns with how different tasks distribute their reasoning over the denoising trajectory. EDIT applies **the same stability rule uniformly**, and its behavior is shaped by the reasoning characteristics of each task.
>
> # Influence of prompt form changes, noise disturbances, or adversarial interventions
> EDIT has not been tested under prompt-format variations, synthetic noise, or adversarial perturbations. The stopping criterion relies on detecting stabilization during inference, and our evaluation focuses on reasoning benchmarks where this setting is most relevant. Extending the study to include controlled perturbations is a constructive suggestion and can be incorporated into follow-up work.

---

### Meta-Review · Area_Chair_XCaV · 2026-01-06

**Summary:**

This paper proposes EDIT, an early-termination rule to accelerate inference for diffusion-style language models by reusing training-time optimizer statistics that are typically discarded. The method compresses aggregated optimizer moments (e.g., AdamW statistics on LoRA-B) into a compact per-block signal, and during inference monitors alignment/stability of token activations across denoising steps. EDIT early-stops when a matched-support KL divergence criterion remains below a threshold for a fixed number of consecutive steps. The paper includes a theoretical argument connecting KL bounds to prediction stability, and reports substantial denoising-step reductions (up to ~68%) with small or no accuracy loss on several reasoning benchmarks.

The reviewers raised several concerns that informed the decision. A central concern is applicability and generality: EDIT requires access to training-time optimizer states/metadata, which may be unavailable for released checkpoints or closed-source models; experiments focus primarily on LoRA-SFT and a single diffusion LLM family (LLaDA-8B), raising questions about robustness across models, sizes, optimizers, adapters, and training recipes. Another major concern is evaluation rigor and baselines: one reviewer argued that the paper initially lacked comparisons to other early-stopping/acceleration methods and that the evaluation potentially confounds gains with task-level hyperparameter tuning, especially since baselines were not similarly tuned for denoising steps. Reviewers also raise concerns that the pipeline’s complex design is insufficiently justified. They argue that comprehensive ablations are required to determine which components are essential, to provide experimental or theoretical justification for each, and to compare against simpler alternatives.

Overall, while some reviewers found the method clear and potentially impactful, the negative assessment highlights remaining doubts about fairness of comparisons, reliance on task-level tuning, breadth of validation, and practical applicability, which leads to an overall recommendation of rejection.

**Reviewer Concerns:**

### Concerns addressed by the rebuttal

* **Generality beyond AdamW + LoRA-SFT (KgfN):**
  The authors report additional experiments switching **AdamW → Lion** (with LoRA) and **LoRA → DoRA** (with AdamW), including accuracy and step statistics on Countdown.

* **End-to-end efficiency metrics (GDvF):**
  The authors provide **wall time per instance, tokens per second, peak GPU memory**, and step reduction for **Countdown@256**.

* **Diagnostic for long-sequence accuracy drop (GDvF):**
  For **GSM8K@512**, the authors state they include a histogram of early-stop steps vs correctness (Figure 8, Section C.3) and report the qualitative pattern that correct predictions stop later while incorrect ones stop earlier.

* **Tolerance to moderate hyperparameter mis-tuning (GDvF, 3GSH, uwNg):**
  The authors state that a **narrow range** of (\delta) and (\Omega) (e.g., (\delta) 0.05–0.1, (\Omega) 6–12) yields consistent behavior across multiple tasks.

* **Baselines vs other acceleration methods (3GSH):**
  The authors add comparisons to **Fast-dLLM** and **Prophet** on **GSM8K@256** and **GPQA@256** under a stated 0-shot setting, and comment on the step count comparison for GSM8K@256.

* **Clarification of design choices and “what distribution is over” (uwNg):**
  The authors clarify the probability distribution is defined over **visible (unmasked) tokens** at a step, and KL is computed on the **intersection** of visible-token sets across consecutive steps after renormalization.

* **Addressing some design-choice critiques (uwNg):**
  The authors report an ablation using **momentum only** vs “combined update” and claim combined update is more stable and accurate on Countdown.

* **Presentation issues (3GSH):**
  The authors state they enlarged tables and increased figure font sizes and removed unintended frames in several figures.

### Concerns still outstanding

* **Practical applicability limited by training-metadata requirement (KgfN, 3GSH):**
  The authors acknowledge that if optimizer states are not preserved, EDIT cannot be applied. This remains a core limitation for settings with only final checkpoints or closed-source models.

* **Breadth of validation across models, sizes, datasets, and hardware (3GSH):**
  The experiments are still described by the reviewer as centered on **LLaDA-8B** and a limited set of tasks/hardware. The rebuttal characterizes broader evaluation as future work but does not add evidence beyond the reported comparisons.

* **Task-level tuning burden and fairness of comparisons (3GSH, uwNg):**
  A key criticism is that ((\delta,\Omega,\tau_{blk})) are **selected per task via validation**, which introduces tuning overhead and potential brittleness. The authors argue a narrow parameter range works broadly and that EDIT can run with defaults, but the method still uses **task-level selection** in the reported results.
  Additionally, uwNg questions whether baselines should also be tuned for number of diffusion steps and notes that EDIT’s step count can fall below the smallest baseline step budget tested. The authors respond that baselines use default schedules and are not retuned, but this does not resolve the reviewer’s fairness concern (i.e., asymmetry in tuning/search over step counts).

* **Stronger comparative evaluation vs simple early-stopping heuristics (uwNg):**
  uwNg highlights missing comparisons to output-level heuristics (e.g., confidence/entropy/output stability) and other alternative early-stopping ideas, and expresses concern that accuracy improvements could reflect tuning effects rather than principled stopping. The rebuttal explains why EDIT uses training-aligned signals but does not provide the requested empirical comparisons to those alternative early-stopping baselines.

* **Concerns about theory/intuition being ad hoc or insufficient (3GSH, uwNg):**
  While the authors provide intuition for the optimizer-derived vector and the stability criterion, reviewers’ concerns about the theory being “ad-hoc” and the need for deeper rigor/alternative scoping (e.g., oracle optimal steps, maximum attainable speedups, broader ablations of design choices) are only partially addressed.

**Reviewer Scores:**

* **Reviewer KgfN (score: 6):**
  Likely unchanged. The rebuttal directly addresses two key questions (other optimizers/adapters; accuracy dips explanation) with added experiments and clarifications, but the reviewer already indicated “would not mind if rejected,” and the metadata-availability limitation remains.

* **Reviewer GDvF (score: 6):**
  Likely unchanged. The rebuttal provides exactly what was requested: end-to-end system metrics and a GSM8K@512 diagnostic histogram, plus a statement about hyperparameter tolerance.

* **Reviewer 3GSH (score: 4):**
  Likely slightly higher or unchanged. The rebuttal addresses several major complaints: adds comparisons to Fast-dLLM/Prophet, clarifies intuition, discusses domain shift qualitatively, and commits presentation fixes. However, the reviewer’s core concerns about limited scope (single model family), task-tuned thresholds, and perceived ad-hoc theory are only partially resolved.

* **Reviewer uwNg (score: 2):**
  Likely unchanged. Although the rebuttal answers specific technical questions (distribution support, ablation on momentum-only, clarification that Q/K/V across layers are used, and baseline comparisons added elsewhere), uwNg’s principal objections center on lack of rigorous scoping and alternative baselines, and especially the fairness issue around task-level tuning and step-budget comparisons; the rebuttal does not provide the broader early-stopping heuristic comparisons or the baseline step-vs-accuracy curves requested in spirit.

Taken together, while two reviewers are moderately positive and the rebuttal meaningfully improves completeness (metrics, diagnostics, some comparisons), the remaining concerns, particularly around **task-level tuning/fairness, limited robustness testing, and metadata dependence**, is considered a major drawback and prevents the paper from reaching the overall acceptance threshold.

---

### Decision · Program_Chairs · 2026-01-26

Reject